# BETTER INSTRUCTION-FOLLOWING THROUGH MINIMUM BAYES RISK

**Ian Wu**[1] **Patrick Fernandes**[2,3,4] **Amanda Bertsch**[2] **Seungone Kim**[2] **Sina Pakazad**[1]
**Graham Neubig**[2]

[1]C3 AI  [2]Carnegie Mellon University  [3]Instituto de Telecomunicações
[4]Instituto Superior Técnico, University of Lisbon
{ian.wu, sina.pakazad}@c3.ai {pfernand, gneubig}@cs.cmu.edu
{abertsch, seungonk}@andrew.cmu.edu

## ABSTRACT

General-purpose LLM judges capable of human-level evaluation provide not only a scalable and accurate way of evaluating instruction-following LLMs but also new avenues for supervising and improving their performance. One promising way of leveraging LLM judges for supervision is through *Minimum Bayes Risk* (MBR) decoding, which uses a reference-based evaluator to select a high-quality output from amongst a set of candidate outputs. In the first part of this work, we explore using MBR decoding as a method for improving the test-time performance of instruction-following LLMs. We find that MBR decoding with reference-based LLM judges substantially improves over greedy decoding, best-of-N decoding with reference-free judges and MBR decoding with lexical and embedding-based metrics on AlpacaEval and MT-Bench. These gains are consistent across LLMs with up to 70B parameters, demonstrating that smaller LLM judges can be used to supervise much larger LLMs. Then, seeking to retain the improvements from MBR decoding while mitigating additional test-time costs, we explore iterative self-training on MBR-decoded outputs. We find that self-training using Direct Preference Optimisation leads to significant performance gains, such that the self-trained models with greedy decoding generally match and sometimes exceed the performance of their base models with MBR decoding.

## 1 INTRODUCTION

Instruction-following large language models (LLMs) (Chung et al., 2022; Wei et al., 2022) have shown remarkable potential as generalist problem-solvers, prompting extensive efforts to improve their performance. One task that has seen tremendous progress due to LLMs is the evaluation of text generation itself. Recent works find that "LLM-as-a-Judge" frameworks (Zheng et al., 2023; Li et al., 2023; Dubois et al., 2024a) demonstrate strong correlation with human evaluations and significantly outperform lexical (Lin, 2004b; Papineni et al., 2002) and embedding-based methods (Zhang et al.; Yuan et al., 2021; Qin et al., 2023) across a wide range of instruction-following tasks.

While the use of LLM judges has largely focused on the evaluation of outputs, LLM judges can also provide a way to supervise the generations of other LLMs. This generally involves using the judge as a *reference-free* evaluator to score candidate outputs produced by the LLM and then selecting the highest-scoring candidate as the final output in what is known as best-of-N (BoN) decoding (Song et al., 2024). However, prior works find that LLM judges, including powerful proprietary LLMs such as GPT-4, significantly underperform when no human-curated reference answer is available (Ye et al., 2024; Zheng et al., 2023). In contrast, *reference-based* evaluation, where a human-curated reference answer is available, shows significantly higher correlation with human evaluations of outputs. This poses a chicken-and-egg problem: how can we leverage reference-based LLM judges for test time generation if no human references are available?

*Minimum Bayes Risk (MBR) decoding* (Bickel & Doksum, 1977) provides a way of overcoming this problem. In place of the inaccessible human references, MBR decoding leverages other candidate

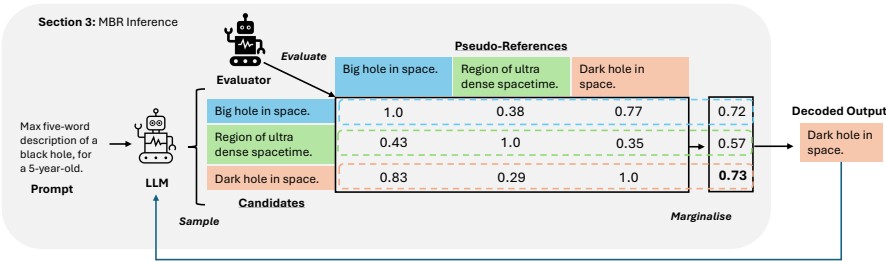

Figure 1: **Illustration of MBR decoding**. Multiple candidates are first sampled from an LLM. Then, each candidate is evaluated against all other candidates (pseudo-references) using a reference-based evaluator. The pseudo-references are marginalised to produce final scores, and the candidate with the highest score is selected.

outputs as *pseudo-references*, and uses the evaluator, also known as the *utility metric*, to conduct reference-based evaluation of all candidate outputs against all pseudo-references. The final output is then chosen as the candidate output with the highest average score: see Figure 1.

In this work, we explore whether MBR decoding using LLM judges as utility metrics can be used to enhance instruction-following LLMs. We divide our work into two main parts. First, inspired by the effectiveness of scaling inference time compute (Welleck et al., 2024; Snell et al., 2024), we investigate whether MBR decoding with LLM judges can improve the performance of instruction-following LLMs during test time (denoted as "MBR inference") (Section 3). Second, following recent works demonstrating that iterative self-training can improve LLM performance (Xu et al., 2023; Chen et al., 2024; Yuan et al., 2024a; Wu et al., 2024), we examine whether MBR decoding with LLM judges can be used to select high-quality model outputs for use in subsequent iterations of self-training (denoted as "MBR distillation") (Section 4). This both provides a way of training models without access to external labels and also allows us to mitigate the inference-time costs associated with MBR inference.

From our MBR inference experiments, we find that MBR decoding with LLM judge Prometheus-2-7B (Kim et al., 2024) improves performance by +3.6% on AlpacaEval and +0.28 on MT-Bench on average across five LLMs relative to greedy decoding. Notably, Llama-2-7b with Prometheus MBR decoding outperforms Llama-2-13b with greedy decoding on MT-Bench, while Prometheus MBR decoding with Llama-2-13b outperforms Llama-2-70b with greedy decoding on AlpacaEval 2.0. Gains persist even for large 70B models, demonstrating that small LLM judges can supervise larger LLMs through MBR decoding. We also compare MBR decoding against other methods that use LLM judges for supervision. We show that Prometheus MBR decoding is far more effective than MBR decoding with word match-based metrics (e.g. ROUGE) or semantic similarity-based metrics (e.g. BERTScore). Comparing MBR to BoN decoding, we find that MBR decoding consistently outperforms BoN decoding across multiple LLMs and LLM judges, and that the gains from MBR decoding increase as the supervising LLM judge increases in size and ability.

From our MBR distillation experiments, we find that self-training with Direct Preference Optimisation (DPO) (Rafailov et al., 2024) on preference pairs selected using MBR decoding (Yang et al., 2024) with Prometheus-2-7B substantially improves greedy decoding performance. For instance, MBR self-trained Llama-2-13b improves by +7.1% on AlpacaEval 2.0 and +0.90 on MT-Bench relative to its baseline SFT counterpart when evaluated using only greedy decoding, far surpassing the corresponding gains from BoN self-training. We also find that MBR self-trained models evaluated with greedy decoding generally match and sometimes exceed the performance of their base models evaluated with MBR decoding, thereby demonstrating that MBR distillation is an effective way of mitigating the inference-time costs of MBR decoding while retaining improved performance.

## 2 BACKGROUND

Language models are *autoregressive* probabilistic models; i.e., the probability of a token $y_i$ depends on prompt $x$ and all previous tokens in the sequence:

$$p(y|x) = \prod_{i=1}^{T} p(y_i|y_{i-1}, \ldots, y_1, x). \tag{1}$$

During inference, outputs are typically obtained either using *maximum a-posteriori*-based decoding methods that attempt to maximise probability, such as greedy decoding ($y_i = \arg\max_{y_i} p(y_i|y_{<i}, x)$) and beam search (Graves, 2012), or by tokenwise sampling from the distribution ($y_i \sim p(y_i|y_{<i}, x)$). Both rely on the model's distribution as indicative of output quality.

Alternatively, we can first obtain a *hypothesis set* $\mathcal{H}_{\text{hyp}}$ comprising $N_{\text{cand}}$ candidate outputs from the model (for example, by sampling multiple times), and then select the final output from $\mathcal{H}_{\text{hyp}}$ based on some external criteria. For example, given some reference-free evaluator $u$ (e.g. an LLM judge), best-of-N (BoN) decoding selects the output $\hat{y} \in \mathcal{H}_{\text{hyp}}$ such that

$$\hat{y} = \arg\max_{y \in \mathcal{H}_{\text{hyp}}} u(y). \tag{2}$$

As reference-free estimation of output quality can be a difficult problem, MBR decoding replaces the reference-free evaluator with a reference-based evaluator $u(y, y^\star)$ (e.g. a reference-based LLM judge) that evaluates candidate $y$ relative to a reference $y^\star$.[1] In the MBR literature, this evaluator is known as a *utility metric* (Freitag et al., 2022; Fernandes et al., 2022; Finkelstein et al., 2024). MBR decoding selects the final output $\hat{y}$ that maximises *expected utility* under the model distribution:

$$\hat{y} = \arg\max_{y \in \mathcal{H}_{\text{hyp}}} \quad \underbrace{\mathbb{E}_{y* \sim p(y|x)}[u(y, y^*)]}_{\approx \frac{1}{N_{\text{cand}}} \sum_{j=1}^{N_{\text{cand}}} u(y, y^{(j)})} \quad , \tag{3}$$

where the expectation is approximated as a Monte Carlo sum using model samples $y^{(1)}, \ldots, y^{(N_{\text{cand}})} \sim p(y|x)$. In practice, this amounts to computing the utility of each candidate in $\mathcal{H}_{\text{hyp}}$ using all other candidates as (pseudo-)references, and then selecting the candidate with the highest average utility as the final output[2] - see Appendix I.1 for an algorithmic description of MBR decoding. The MBR-decoded output can therefore be interpreted as being the candidate with the highest *"consensus"* utility as measured by the utility metric, as it achieves the highest average utility when evaluated against all other candidate outputs. It is therefore crucial to choose a reference-based metric that is a good proxy for human preferences as our utility function, as this ensures that a "high-consensus" output corresponds to a "high-quality" output.

## 3 MBR INFERENCE

In this experiment, we investigate using MBR decoding with LLM judge utility metrics to improve instruction-following LLMs at test time.

### 3.1 EXPERIMENTAL SETUP

#### 3.1.1 MODELS AND GENERATION PARAMETERS

We use the chat and instruct variants of the Llama-2 (Touvron et al., 2023b) and Llama-3 (Dubey et al., 2024) models in this experiment. All models have undergone prior SFT and demonstrate strong instruction-following and conversation abilities. We generate $N_{\text{cand}} = 30$ candidates using temperature sampling with $t = 0.3$ for all MBR decoding experiments unless otherwise specified.

#### 3.1.2 MBR UTILITY METRICS

**LLM judge** We choose **Prometheus-2-7B** (Kim et al., 2024) as our representative reference-based LLM judge. Prometheus is a specialist judge model finetuned from Mistral-7b (Jiang et al., 2023) that correlates strongly with human judges and GPT-4. It takes as inputs a task prompt, a scoring rubric (see Appendix C), a candidate and a reference, and outputs an explanation of its judgement followed by a score from 1 to 5, which we interpret as a utility score. Crucially, Prometheus can also act as a reference-free judge by simply omitting the reference from its input. This allows us to directly compare MBR with BoN decoding using the same LLM judge utility metric.

---

[1]Certain evaluators (e.g. LLM judges) are "task-aware", and take prompt $x$ as an input when performing evaluation. Such utility metrics can then be written as $u(y; x)$ and $u(y, y^\star; x)$.

[2]The expectation can also be computed over a separate set of model outputs known as the *evidence set* (Eikema & Aziz, 2020; Bertsch et al., 2023). We do not explore this setting in our work.

We compare using LLM judges for MBR decoding with three other classes of utility metrics:

**ROUGE**   ROUGE (Lin, 2004a) is a word-level F-measure designed for measuring summarisation and machine translation performance (Lin, 2004a). We use ROUGE-1 in our main experiments but include results for other ROUGE variants in Appendix A.3.

**BERTScore**   BERTScore is a neural evaluation metric that computes the token-level contextual similarity between a candidate and a reference. Like ROUGE, BERTScore is not task-aware. As our model outputs may be longer than 512 tokens, we use **Longformer-large** (Beltagy et al., 2020) as our BERTScore model, which supports inputs up to 4094 tokens long.

**Dense embedders**   Dense embedders generate contextual embeddings of text passages for use in downstream tasks. One such task is measuring the level of semantic similarity between two text passages (Agirre et al., 2012). This task is directly relevant to MBR decoding, as we can treat pairs of candidates as text passages and their similarity score as the utility. To the best of our knowledge, using dense embedders as a utility metric for MBR decoding has never been explored before. In our work, we use the instruction-following embedder **SFR-Embedder-2_R** (Meng et al., 2024) as our representative dense embedder. We include results for two other dense embedders in Appendix A.3.

### 3.1.3   BASELINES

In addition to **greedy decoding** and **beam search** (BS) with $k = 10$ beams, we also experiment with **LONGEST decoding**, where we select the longest candidate from the hypothesis set (as measured in characters) as the final output, and **best-of-N (BoN) decoding**. We generate $N_{cand} = 30$ candidates using temperature sampling with $t = 0.3$ for both longest and BoN decoding. See Appendices A.2 and A.3 for comparisons with additional baselines.

### 3.1.4   EVALUATION

**AlpacaEval 2.0**   AlpacaEval (Li et al., 2023) is an LLM-based evaluation metric. It consists of an 805-sample, highly diverse single-turn instruction-following conversational dataset and an associated evaluation framework. In AlpacaEval 2.0 (Dubois et al., 2024b), evaluation is conducted by performing head-to-head comparison of candidate answers against GPT-4-generated answers facilitated by a judge LLM. The judge model is prompted to output a single token representing its choice of winner, with the log-probabilities of the token used to compute a weighted win rate. In addition to standard win rates, AlpacaEval 2.0 also provides length-controlled (LC) win rates, which are debiased versions of the standard win rates that control for the length of the outputs. Both the AlpacaEval standard and LC evaluation demonstrate strong correlation with human judgements.

**MT-Bench**   MT-Bench (Zheng et al., 2023) is an 80-sample, two-turn instruction-following conversational dataset. It can be evaluated using either head-to-head comparison or direct assessment with an LLM judge. In the direct assessment setting, the judge LLM is prompted to generate an explanation followed by a score between 1 and 10, with no reference answer used. MT-Bench with GPT-4-judge matches crowdsourced human preferences well, achieving over 80% agreement, which is the same level of agreement between human evaluators (Zheng et al., 2023).

We use GPT-4o (OpenAI et al., 2024) as the LLM judge for both AlpacaEval 2.0 and MT-Bench. For AlpacaEval, we report LC win rates unless otherwise stated. For MT-Bench, we use direct assessment for all experiments. See Appendix B for further details on our evaluation strategy, and Appendix H for human study findings verifying the alignment of our automatic LLM evaluation results with human judgements.

## 3.2   EXPERIMENTAL RESULTS

Our main experimental results are documented in Tables 1 and 2.

**Prometheus MBR decoding provides significant and consistent gains**   The gains associated with Prometheus MBR decoding are significantly larger than those associated with other utility metrics, yielding an average improvement of +3.6% on AlpacaEval 2.0 and +0.28 on MT-Bench. For comparison, the performance gap between Llama-2-7b and Llama-2-13b with greedy decoding is +4.6% on AlpacaEval 2.0 and +0.18 on MT-Bench, while the corresponding gap between Llama-2-13b and Llama-2-70b is +3.8% and +0.60. Notably, Llama-2-7b with Prometheus MBR

|  | **2-7B** | **2-13B** | **2-70B** | **3-8B** | **3-70B** | **Avg. $\Delta$** |
|---|---|---|---|---|---|---|
| Greedy | 14.4 | 19.0 | 22.8 | 34.4 | 42.7 | 0 |
| BS | 14.8 | 18.2 | 21.5 | 33.9 | 42.4 | -0.50 |
| Longest | 10.5 | 15.2 | 19.8 | 29.8 | 40.4 | -3.51 |
| Prometheus BoN | 16.4 | 20.8 | 25.0 | 35.5 | 44.3 | 1.74 |
| ROUGE MBR | 16.2 | 20.0 | 24.7 | 35.4 | 43.7 | 1.33 |
| BERTScore MBR | 16.2 | 20.5 | 24.4 | 35.7 | 44.0 | 1.50 |
| SFR-Embedder MBR | 12.1 | 16.6 | 22.2 | 32.5 | 42.8 | -1.42 |
| Prometheus MBR | **17.7** | **23.4** | **26.2** | **37.9** | **46.0** | **3.62** |

Table 1: AlpacaEval 2.0 win rates (%) for various models and decoding strategies, along with the average win rate differences compared to greedy decoding across all models (denoted as **Avg.** $\Delta$). MBR decoding with Prometheus consistently outperforms all baseline methods and other MBR decoding methods.

|  | **2-7B** | **2-13B** | **2-70B** | **3-8B** | **3-70B** | **Avg. $\Delta$** |
|---|---|---|---|---|---|---|
| Greedy | 5.72 | 5.90 | 6.50 | 7.54 | 8.29 | 0 |
| BS | 5.58 | 5.95 | 6.49 | 7.30 | 8.20 | -0.09 |
| Longest | 5.67 | 6.03 | 6.59 | 7.22 | 8.22 | -0.04 |
| Prometheus BoN | 5.77 | 6.08 | 6.65 | 7.66 | 8.42 | 0.13 |
| ROUGE MBR | 5.78 | 6.11 | 6.68 | 7.63 | 8.31 | 0.11 |
| BERTScore MBR | 5.68 | 6.02 | 6.72 | 7.52 | 8.42 | 0.08 |
| SFR-Embedder MBR | 5.73 | 6.04 | 6.54 | 7.45 | 8.33 | 0.03 |
| Prometheus MBR | **6.10** | **6.26** | **6.79** | **7.69** | **8.50** | **0.28** |

Table 2: MT-Bench scores for various models and decoding strategies, along with the average score differences compared to greedy decoding across all models (denoted as **Avg.** $\Delta$). MBR decoding with Prometheus consistently outperforms all baseline methods and other MBR decoding methods.

decoding outperforms Llama-2-13b with greedy decoding on MT-Bench, while Prometheus MBR decoding with Llama-2-13b outperforms Llama-2-70b - a model over five times bigger - with greedy decoding on AlpacaEval 2.0. We also find that Prometheus MBR decoding yields larger gains than Prometheus BoN decoding; we explore this further in Section 3.3.1.

We also highlight that the performance gains associated with Prometheus MBR decoding are significant across models of all sizes, even for much larger models such as Llama-3-70b. This scaling property suggests that small judge models can still be used to supervise much larger models.

**ROUGE and BERTScore MBR decoding provide small but consistent gains**    ROUGE and BERTScore MBR decoding improve average performance relative to greedy decoding by +1.3% and +1.5% on AlpacaEval 2.0 and by +0.11 and +0.08 on MT-Bench respectively. This benefit is present for all models. This improvement suggests that selecting outputs without awareness of the task and using only word- or token-level measures of consistency can still yield meaningful improvements even in the instruction-following setting.

**SFR-Embedder MBR decoding fails to yield consistent gains**    SFR-Embedder MBR decoding reduces performance relative to greedy decoding by -1.4% on AlpacaEval 2.0 while improving performance by +0.03 on MT-Bench on average. We hypothesise that embedder models, which are trained to distinguish at a high level between text passages, cannot to detect nuanced differences between semantically-similar outputs. We also note that embedder MBR decoding generally selects for longer outputs, which may explain the discrepancy between its performance on AlpacaEval 2.0 (which is length-controlled) and MT-Bench. See Appendix A.4 for analysis on the generation lengths of various decoding strategies.

**Beam search and LONGEST decoding degrade performance**    Beam search and LONGEST decoding reduce performance relative to greedy decoding by -0.5% and -3.5% on AlpacaEval 2.0 and -0.09 and -0.04 on MT-Bench respectively. The poor performance of beam search further underscores the idea that optimising for output probability alone is not enough to improve output quality.

## 3.3    ANALYSIS OF PROMETHEUS MBR DECODING

Given the promise shown by Prometheus MBR decoding, we conduct additional experiments to better understand its properties.

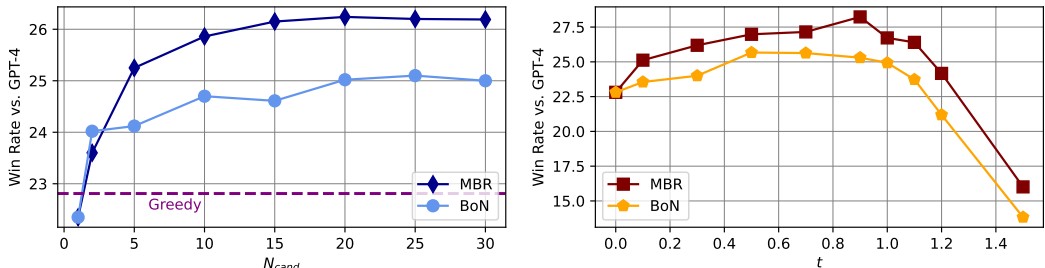

Figure 2: AlpacaEval 2.0 win rates (%) for Llama-2-70b with varying hypothesis set size $N_{\text{cand}}$ **(left)** and generation temperature $t$ **(right)** values for Prometheus MBR and BoN decoding. Performance for both methods initially increases with $N_{\text{cand}}$ and plateaus at around $N_{\text{cand}} = 20$. Performance also initially increases with $t$, but drops rapidly after $t = 1.0$.

**Prometheus MBR decoding performance vs. $t$ and $N_{\text{cand}}$** We plot AlpacaEval 2.0 win rates as a function of $N_{\text{cand}}$ and $t$ in Figure 2 for Llama-2-70b with Prometheus MBR and BoN decoding. We find that performance initially increases with $N_{\text{cand}}$ but plateaus at around $N_{\text{cand}} = 20$, suggesting that expanding the size of the hypothesis set beyond this yields little benefit. Performance also initially increases with $t$, highlighting the benefits of increased candidate diversity, although it rapidly degrades at high temperatures as the individual candidate outputs decline in quality.

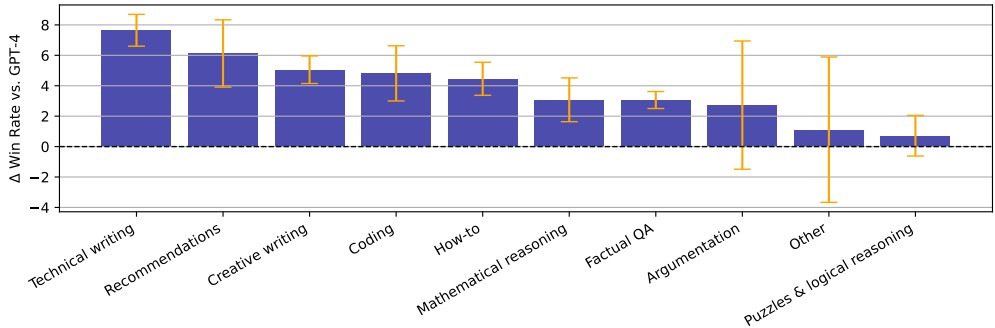

Figure 3: Difference in AlpacaEval 2.0 win rates (%) between Prometheus MBR decoding and greedy decoding averaged over all five LLMs and broken down by question category. A positive value indicates that MBR decoding outperforms greedy decoding on the given category. Orange bars represent the standard error. We find that Prometheus MBR decoding improves performance across a wide range of question categories.

**Prometheus MBR decoding performance by question category** We classified questions from the AlpacaEval dataset into one of ten categories using GPT-4o (see Appendix D for details), and then computed the differences in win rates between Prometheus MBR decoding and greedy decoding by question category, averaged over all five LLMs. We find that MBR decoding improves output quality across most question categories. These include reasoning-based categories such as coding and mathematical reasoning, although the largest improvements are seen across writing-based categories such as technical, recommendations and creative writing. We hypothesise that this discrepancy arises due to (1) the higher bar for correctness associated with reasoning tasks which limits the number of good answers that can be found amongst candidate outputs; and (2) limitations of existing utility functions, which may struggle to handle difficult reasoning tasks.

### 3.3.1 FURTHER COMPARISONS WITH BEST-OF-N DECODING

As BoN decoding can also leverage LLM judges as a utility metric, we conduct additional experiments to compare its performance against MBR decoding. We compare BoN and MBR decoding for five different LLM judges on MT-Bench and report the results in Table 3. In addition to Prometheus-2-7B, we also evaluate its larger sibling **Prometheus-2-8x7B**, as well as **JudgeLM-7b** and **JudgeLM-33b** (Zhu et al., 2023c), which are two judge models finetuned from LLaMA models (Touvron et al., 2023a). We also assess **Llama-3-8b-Instruct** and **Llama-3-70b-Instruct** as zero-

|  | 2-7B | 2-13B | 2-70B | 3-8B | 3-70B | Avg. $\Delta$ |
|---|---|---|---|---|---|---|
| Greedy | 5.72 | 5.90 | 6.50 | 7.54 | 8.29 | 0 |
| Prometheus-2-7B-BoN | 5.77 | 6.08 | 6.65 | 7.66 | 8.42 | 0.13 |
| Prometheus-2-7B-MBR | 6.10 | 6.26 | 6.79 | 7.69 | 8.50 | 0.28 |
| Prometheus-2-8x7B-BoN | 6.01 | 6.17 | 6.80 | 7.75 | 8.41 | 0.24 |
| Prometheus-2-8x7B-MBR | **6.26** | 6.32 | 6.87 | 7.79 | **8.64** | 0.39 |
| JudgeLM-7b-BoN | 5.63 | 5.95 | 6.69 | 7.37 | 8.26 | -0.01 |
| JudgeLM-7b-MBR | 6.00 | 6.11 | 6.79 | 7.69 | 8.44 | 0.22 |
| JudgeLM-33b-BoN | 5.68 | 6.03 | 6.58 | 7.37 | 8.35 | 0.01 |
| JudgeLM-33b-MBR | 5.94 | 6.27 | 6.88 | **7.92** | 8.50 | 0.31 |
| Llama-3-8b-Instruct-BoN | 5.83 | 6.05 | 6.61 | 7.60 | 8.38 | 0.10 |
| Llama-3-8b-Instruct-MBR | 5.96 | 6.28 | 6.84 | 7.80 | 8.47 | 0.28 |
| Llama-3-70b-Instruct-BoN | 5.77 | 6.16 | 6.57 | 7.39 | 8.35 | 0.06 |
| Llama-3-70b-Instruct-MBR | 6.22 | **6.43** | **6.94** | 7.87 | 8.52 | **0.41** |

Table 3: MT-Bench scores for BoN and MBR decoding with various judge LLMs as utility metrics, along with the average score differences compared to greedy decoding across all models (denoted **Avg.** $\Delta$). MBR decoding consistently outperforms BoN decoding across all comparable utility metrics.

shot judges for MBR decoding (see Appendix E for our prompts). All chosen judges can act as both reference-free and reference-based judges, allowing us to compare MBR and BoN decoding fairly.[3]

We find that MBR decoding consistently outperforms BoN decoding across all selected judge models. This difference is especially large for the JudgeLM models and for Llama-3-70b-Instruct, where BoN fails to significantly improve on greedy decoding. One explanation for this discrepancy is that our LLM judges are insufficiently good at reference-free evaluation for BoN decoding to be effective. This idea is supported by prior studies comparing reference-free and reference-based evaluation, which consistently show that reference-free methods tend to underperform, even when using strong judge models like GPT-4 (Ye et al., 2024; Kim et al., 2024; Zheng et al., 2023). Another explanation is that MBR decoding provides a smoothing effect that arises from our use of expected utility in place of utility point estimates for output selection, tying back to our hypothesis that selecting "high-consensus" outputs yields significant benefit. This averaging process reduces the impact of individual mistakes made by the imperfect LLM judge, thereby providing for a more stable and reliable measure of quality. We leave further exploration of these ideas to future work.

Notably, in Table 3, MBR performance improves by scaling the size of the LLM judge, with Prometheus-2-8x7B outperforming Prometheus-2-7B, JudgeLM-33b outperforming JudgeLM-7b, and Llama-3-70b-Instruct outperforming Llama-3-8b-Instruct. This suggests that improving the LLM judge utility metric directly improves MBR decoding performance and that MBR decoding will benefit as newer and better LLM judges are developed.

## 4 MBR DISTILLATION

Our results so far demonstrate the potential of MBR decoding to significantly improve the test-time performance of instruction-following models, but this comes at the cost of substantial inference-time compute costs due to the linear cost for generating $N_{cand}$ candidate outputs and the quadratic cost for computing utility across these candidates. To mitigate this, we explore distilling MBR-decoded outputs back into the model itself and aim to obtain MBR decoding-level (or better) performance without needing to perform MBR decoding at test time.

### 4.1 EXPERIMENTAL SETUP

#### 4.1.1 TRAINING AN SFT MODEL

We start by performing SFT on the base Llama-2-7b and Llama-2-13b models. This is necessary to instil instruction-following behaviour in the models so that they can be used to generate instruction-following self-training data. We choose not to use the official chat variants of these models as we

---

[3]Because sequence-classifier reward models (Stiennon et al., 2022) do not support reference-based evaluation, it is not possible to fairly compare BoN decoding with these methods to MBR. We therefore do not discuss this in the main text and report our findings in Appendix F instead.

wish to retain control over the training procedure and avoid inheriting any biases introduced through prior finetuning and alignment. We use 3000 random samples from UltraChat (Ding et al., 2023) for SFT. UltraChat is a diverse conversational instruction-following dataset created using GPT-3.5-Turbo. Each sample consists of multi-turn prompts and responses, although we only take the first turn of each sample in order to simplify experimentation. We designate our SFT models as *sft*.

### 4.1.2 ITERATIVE DPO ON MBR-DECODED OUTPUTS

Having obtained SFT models, we now conduct DPO to improve the models on their own MBR-decoded outputs, an approach first proposed by Yang et al. (2024) for improving machine translation. We start by randomly drawing a further 3000 prompts from UltraChat (excluding the samples that have already been selected for SFT). Next, we generate $N_{\text{cand}} = 12$ candidate outputs from our *sft* models using these prompts. We use a smaller $N_{\text{cand}}$ than for MBR inference to balance performance and compute cost, as we know from Figure 2 that using an $N_{\text{cand}}$ value above 10 already yields significant gains. Following Yang et al. (2024), we then score the candidate outputs using a utility metric and form preference pairs from the highest-scoring and lowest-scoring outputs. This preference pair dataset is then used for DPO on the *sft* models, yielding *dpo*-MBR-1. We extend upon Yang et al. (2024) by iteratively repeating this process twice more, each time using the latest *dpo* models as the base model paired with a fresh set of 3000 prompts, yielding the models *dpo*-MBR-2 and *dpo*-MBR-3. See Appendix K for a summary of our SFT and DPO hyperparameters, Appendix G.5 for experimental results from using another preference pair selection strategy, and Appendix I.2 for mathematical and algorithmic overviews of MBR distillation.

### 4.1.3 UTILITY METRICS AND EVALUATION

We use Prometheus-2-7B as our utility metric, although we also try MBR self-training with ROUGE and Llama-3-8b-Instruct as the utility metrics (Appendix G.3 and G.4). We compare our *dpo* models with greedy decoding against the *sft* models with greedy decoding, beam search and MBR decoding. For MBR decoding, we use $N_{\text{cand}} = 30$ and $t = 0.3$ with Prometheus-2-7B as the utility metric. We also baseline against models trained with SFT on 12000 UltraChat samples (*sft*-full). Finally, we experiment with BoN self-training, again using Prometheus-2-7B as the utility metric and following the same procedure as for MBR self-training, which yields the models *dpo*-BoN-1, *dpo*-BoN-2 and *dpo*-BoN-3.

We evaluate our trained models using greedy decoding on AlpacaEval 2.0, once again reporting length-controlled win rates vs. GPT-4, and MT-Bench. As we only train our models to engage in single-turn conversations we evaluate only on the first turn of MT-Bench. We report additional evaluation results in the Appendix, including head-to-head results between various self-trained models and *sft* with greedy decoding (Appendix G.1), evaluation results on a selection of popular NLP benchmarks (Appendix G.2), and human study results (Appendix H).

## 4.2 RESULTS

| | AlpacaEval 2.0 | | MT-Bench | |
|---|---|---|---|---|
| | **7B** | **13B** | **7B** | **13B** |
| *sft* w. Greedy | 5.18 | 8.24 | 5.43 | 5.85 |
| *sft* w. MBR | **9.99** | 13.6 | 5.78 | 6.31 |
| *sft*-full | 6.35 | 9.40 | 5.55 | 6.26 |
| *dpo*-1-BoN | 5.78 | 10.3 | 5.78 | 6.08 |
| *dpo*-2-BoN | 6.22 | 11.2 | 5.91 | 6.41 |
| *dpo*-3-BoN | 6.40 | 12.8 | 5.88 | 6.56 |
| *dpo*-1-MBR | 5.68 | 10.8 | 5.78 | 6.48 |
| *dpo*-2-MBR | 7.22 | 13.9 | 6.11 | 6.73 |
| *dpo*-3-MBR | 8.86 | **15.3** | **6.14** | **6.75** |

| | AlpacaEval 2.0 | MT-Bench |
|---|---|---|
| *sft*-1-MBR | 5.52 | 5.48 |
| *sft*-2-MBR | 6.75 | 5.43 |
| *sft*-3-MBR | 6.48 | 5.51 |

Table 4: **(Left)** AlpacaEval 2.0 win rates (%) and MT-Bench scores for models self-trained using DPO. After three rounds of training, the self-trained models consistently outperform their BoN counterparts and SFT baselines. **(Top)** AlpacaEval 2.0 win rates (%) and MT-Bench scores for models self-trained using SFT. Self-training with SFT yields substantially worse results than self-training with DPO.

**DPO self-training significantly improves model performance** We report the results of our self-training experiment in the left subtable of Table 4. We find that three rounds of MBR self-training with DPO significantly improves model performance, with the 7B *dpo*-3-MBR model outperforming

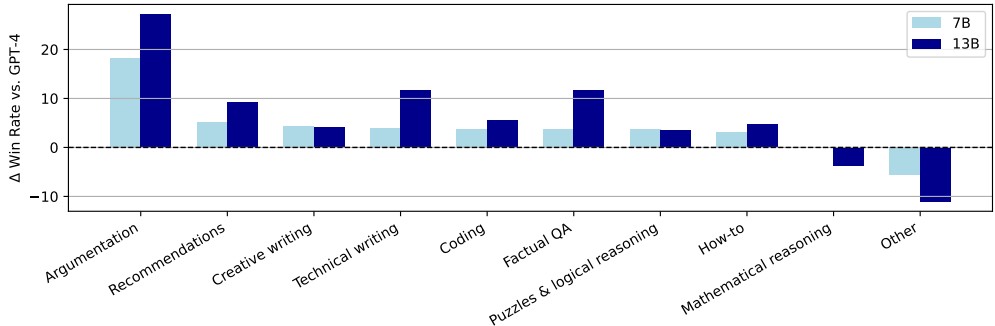

Figure 4: Differences in AlpacaEval 2.0 win rates (%) between *dpo*-3-MBR models and their respective *sft* with greedy decoding baselines on different question categories. The largest improvements are seen in open-ended writing tasks, with less improvement on reasoning-focussed tasks (e.g. mathematical reasoning and coding).

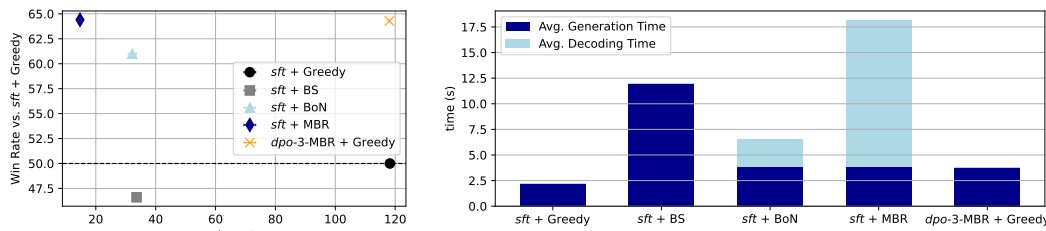

Figure 5: **(Left)** AlpacaEval 2.0 win rate (%) vs. *sft* with greedy decoding against generation throughput. *dpo*-3-MBR with greedy decoding matches the performance of *sft* with MBR decoding, but with significantly higher throughput. **(Right)** Average generation and decoding times on the AlpacaEval dataset. The decoding step in MBR decoding takes disproportionately long. This problem can mitigated through MBR self-training.

the 13B *sft* model with greedy decoding on both AlpacaEval 2.0 and MT-Bench. The improvements saturate by the third round of self-training as measured on MT-Bench (6.11 vs. 6.14 (7B) and 6.73 vs. 6.75 (13B) in Table 4), although there appears to be room for further improvement on AlpacaEval 2.0 (7.22 vs. 8.86 (7B) and 13.9 vs. 15.3 (13B) in Table 4). Both *dpo*-3-MBR models outperform their *sft*-full counterparts, which suggests that training on MBR-decoded outputs is more beneficial than SFT on a larger split of UltraChat. The *dpo*-3-MBR models also generally outperform *sft* with MBR decoding, and this is especially prominent for MT-Bench, which suggests that DPO on MBR-decoded outputs enables models to recover and then exceed their MBR-decoding performances. We find that DPO on BoN-decoded outputs also improves model performance, although less so than DPO with MBR-decoded outputs. We attribute this to the relative strength of MBR decoding.

**SFT self-training yields smaller gains than DPO self-training**   We experiment with iterative SFT self-training, using the 7B *sft* model. We document our results in the right subtable of Table 4. We use the same sets of prompts as for DPO and select as our SFT labels the highest-scoring sample as determined by MBR, following Finkelstein et al. (2024). As before, we conduct three rounds of iterative training, yielding *sft*-1-MBR, *sft*-2-MBR and *sft*-3-MBR. We find that SFT training yields significantly less improvement than DPO. This indicates that MBR self-training benefits most from preference learning, where the model learns to contrast its highest- and lowest-quality outputs.

**DPO self-trained model performance by question category**   We repeat the analysis on performance by question category for *dpo*-3-MBR in Figure 4. Self-training improves performance on almost all question categories, with generally larger improvement on writing-based categories and smaller improvement on reasoning-based categories. We attribute this difference to the writing-skewed distribution of question categories in our UltraChat training data (see Appendix G.6).

**Analysis of compute costs**   We illustrate the savings on compute introduced by self-training in Figure 5. We perform inference with various 7B models and decoding strategies on AlpacaEval 2.0, using 2xA100 GPUs and vLLM (Kwon et al., 2023) as our inference engine. We use a generation batch size of 1, a LLM judge utility calculation batch size of 32, and $N_{\text{cand}} = 12$. We find that MBR decoding imposes significant overhead, largely due to the quadratic $\mathcal{O}(N_{\text{cand}}^2)$ cost incurred during

the utility calculation step. This overhead is removed through MBR self-training, which nonetheless retains performance gains. Note that *dpo*-3-MBR generates longer outputs than *sft*, which explains why its average generation time as seen in the right-hand plot of Figure 5 is higher.

## 5    RELATED WORK

**MBR decoding**   MBR decoding has been explored in the context of machine translation using a variety of translation metrics such as COMET (Rei et al., 2020) and BLEURT (Sellam et al., 2020), with promising results (Freitag et al., 2022; 2023; Farinhas et al., 2023; Stanojević & Sima'an, 2014). Prior works (Bertsch et al., 2023; Jinnai et al., 2023) also study MBR decoding for summarisation, using ROUGE and BERTScore as metrics. Suzgun et al. (2022) apply MBR decoding to several tasks, including summarisation, machine translation and three different BIG-Bench tasks (Srivastava et al., 2023). None of these works explore the use of MBR decoding in the more open-ended instruction-following domain, nor do they consider using LLM judges as utility metrics.

**LLM judges**   Based on the strong instruction-following capabilities of LLMs, recent works explore prompting LLMs to judge responses from other LLMs (Li et al., 2023; Zheng et al., 2023). Follow-up works suggest that training on the evaluation traces of strong models may equip smaller models with strong evaluation capabilities (Kim et al., 2023; 2024; Zhu et al., 2023b; Vu et al., 2024; Wang et al., 2024b). These works focus on training LLMs to produce scoring decisions matching those of humans. In our work, instead of viewing evaluation as an end goal, we explore utilising the evaluation capabilities of LLM judges as supervision to improve instruction-following LLMs.

**Inference-time algorithms**   Many inference-time algorithms generate candidate outputs and select a final output based on external criteria. In addition to MBR and BoN decoding, examples include Self-Consistency (Wang et al., 2023), which selects the most self-consistent answers through marginalisation of chain-of-thought reasoning paths and Universal Self-Consistency (USC) (Chen et al., 2023), where the LLM is used to self-select consistent chain-of-thought reasoning paths from amongst many reasoning paths. Kuhn et al. (2023) propose an MBR-esque algorithm that uses dense embedders and clustering to measure semantic uncertainty. Other inference-time algorithms prompt the LLM to perform additional inference steps in a structured manner. Examples include Tree-of-Thoughts (Yao et al., 2023) and Graph-of-Thoughts (Besta et al., 2024), as well as recursive improvement strategies such as Self-Refine (Madaan et al., 2023) and Reflexion (Shinn et al., 2023).

**Self-training** Self-training is a promising avenue for model improvement as it enables training without labelled data. Gulcehre et al. (2023) introduce an algorithm that generates samples from a policy and then updates the policy using offline RL. Yuan et al. (2024b) train models to score their own outputs, and then use these scores to create preference datasets which for distillation. Huang et al. (2022) train models on their own highest-confidence outputs as determined by majority voting. Self-training on MBR-decoded outputs has also been explored for machine translation. Finkelstein et al. (2024) train models with SFT on their own MBR and quality estimation outputs for machine translation and demonstrate that this yields improvements over baseline models. Wang et al. (2024a) use MBR to generate targets for sequence-level distillation, again for machine translation. Yang et al. (2024) are the first to use DPO to upweight the model's own MBR generations, allowing them to recover much of their original MBR performances on translation using only greedy decoding.

## 6    CONCLUSION

In this work, we investigate using LLM judges to supervise other LLMs on instruction-following tasks through MBR decoding, and find that this yields significant and consistent improvements to model performance relative to greedy decoding, beam search and BoN decoding. These benefits persist across a wide range of question categories and are also consistent across models of various sizes, demonstrating that small LLM judges can be used to improve much larger LLMs at inference time. To mitigate the significant inference-time costs associated with MBR decoding, we also explore iterative self-training on MBR-decoded outputs. We find that MBR self-training using DPO, but not SFT, enables models to recover and even exceed their base MBR decoding performance using only greedy decoding. We hope our work further highlights the potential of using LLM judges for supervision and inspires future research into MBR decoding beyond its traditional domains and applications, particularly through the development of new utility metrics.

## REPRODUCIBILITY STATEMENT

In an effort to make our work reproducible, we document all prompts (Appendices E and B), as well as training and inference hyperparameters (Appendix K) used throughout our experiments. We also include version information for all API-based LLMs (Appendix B), and choose to use open-source models (the Llama-2, Llama-3, Prometheus-2 and JudgeLM families) where possible.

## ACKNOWLEDGEMENTS

AB was supported by a grant from the National Science Foundation Graduate Research Fellowship Program under Grant No. DGE2140739. PF was supported by the Portuguese Recovery and Resilience Plan through project C645008882-00000055 (Center for Responsible AI) and by FCT/MECI through national funds and when applicable co-funded EU funds under UID/50008: Instituto de Telecomunicações. Any opinions, findings, and conclusions or recommendations expressed in this material are those of the author(s) and do not necessarily reflect the views of the sponsors.

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

# A    MBR DECODING: ADDITIONAL RESULTS

This section presents additional results from our experiments exploring the use of MBR decoding to improve test-time performance.

## A.1    ADDITIONAL MBR UTILITY METRICS

|                       | Llama-2-7B | Llama-2-70B | Avg. $\Delta$ |
|-----------------------|:----------:|:-----------:|:-------------:|
| Greedy                | 14.4       | 22.8        | 0             |
| ROUGE-1 MBR           | 16.2       | 24.7        | 1.85          |
| SFR-Embedder MBR      | 12.1       | 22.2        | -1.45         |
| Prometheus MBR        | **17.7**   | **26.2**    | **3.35**      |
| ROUGE-2 MBR           | 16.6       | 24.6        | 2.00          |
| ROUGE-L MBR           | 15.5       | 24.7        | 1.50          |
| NVEmbed-Embedder MBR  | 14.1       | 22.1        | -0.50         |
| Nomic-Embedder MBR    | 16.3       | 24.1        | 1.60          |

Table 5: AlpacaEval 2.0 win rates (%) for additional MBR decoding experiments, along with the average win rate differences compared to greedy decoding across all models (denoted **Avg. $\Delta$**).

We experiment with two additional ROUGE variants and two additional dense embedders as utility metrics. The two ROUGE variants are ROUGE-2 and ROUGE-L, which detect bigram overlap and longest co-occurring n-gram overlap respectively. The two dense embedders are NVEmbed-v2 (Lee et al., 2024) and Nomic-Text-v1.5 (Nussbaum et al., 2024), which, like SFR Embedder, are strong, long-context dense embedders that rank highly on the MTEB leaderboard (Muennighoff et al., 2023).

We find that MBR decoding with ROUGE-2 performs slightly better than MBR decoding with ROUGE-1, while MBR decoding with ROUGE-L performs slightly worse. MBR decoding with NVEmbed and Nomic both perform better than MBR decoding with SFR Embedder, with Nomic showing some improvement relative to greedy decoding. The latter result suggests that dense embedders could potentially be used for MBR decoding, although further work is required to understand what properties make for good MBR embedders. Overall, none of our additional utility metrics provide comparable improvements to MBR with Prometheus.

## A.2    COMPARISON WITH UNIVERSAL SELF-CONSISTENCY

We compare MBR decoding to Universal Self-Consistency (USC) (Chen et al., 2023). In USC, $N_{cand}$ outputs are sampled from the LLM and passed directly to the LLM for consistency detection. This entails prompting the LLM to choose the most consistent output. In Chen et al. (2023), the authors demonstrate that USC improves over greedy decoding for mathematical reasoning, code generation, summarisation and question-answering.

The limited context lengths of LLMs poses a significant challenge when using USC, as it requires fitting all $N_{cand} = 30$ samples into a single prompt. In contrast, MBR decoding only requires fitting two outputs into a single prompt as utility is computed pairwise. As our existing choice of models have limited context lengths (4096 tokens for Llama-2, 8192 tokens for Llama-3) and our outputs can be long (up to 1024 tokens), we are unable to assess USC on equal footing with MBR decoding using these models without significantly reducing $N_{cand}$. In order to facilitate a fair comparison, we therefore use the Llama-3.1 models (Dubey et al., 2024) in place of the Llama-2 and Llama-3 models for this experiment. The Llama-3.1 models possess context lengths of 128k, thereby allowing us to fit all $N_{cand}$ samples into a single to prompt as required. Our USC prompt is as follows:

```
You are given a collection of 30 responses to a prompt.  Select
the most consistent response based on majority consensus.  The
most concistent response should be the most representative of
all the responses provided.  You should consider a variety of
factors when evaluating consistency, including content, arguments
and examples employed, style, structure and final answer, if
relevant.  Do not pass judgement on the quality or correctness
of the response.  Consider only consistency.

Provide a short explanation of your choice, followed by your
choice.  Your choice should follow this format:  "Most Consistent
Response:  [[Response ID]]", for example:  "Most Consistent
Response:  [[15]]" if response 15 is the most consistent amongst
all responses.

Responses:  {responses}
```

|  | Llama-3.1-8B | Llama-3.1-70B | Avg. $\Delta$ |
|---|---|---|---|
| Greedy | 34.2 | 42.1 | 0 |
| USC | 37.3 | 43.7 | 2.35 |
| MBR Prometheus | **40.2** | **45.8** | **4.85** |

Table 6: AlpacaEval 2.0 win rates (%) for Llama-3.1 models with greedy decoding, USC and MBR decoding with Prometheus, along with the average win rate differences compared to greedy decoding across all models (denoted **Avg. $\Delta$**).

We document our findings in Table 6. We find while that USC provides some improvements over greedy decoding, this improvement is smaller than the improvement provided by MBR decoding with Prometheus.

### A.3 SAMPLING BASELINES

We conduct additional baseline experiments with top-$p$ (Holtzman et al., 2020) and top-$k$ (Fan et al., 2018) sampling. We use these sampling methods to directly obtain a final output, and do not explore replacing standard temperature sampling with these sampling strategies for generating the hypothesis set - we leave this to future work.

|  | Llama-2-7B | Llama-2-70B | Avg. $\Delta$ |
|---|---|---|---|
| Greedy | 14.4 | 22.8 | 0 |
| ROUGE-1 MBR | 16.2 | 24.7 | 1.85 |
| Prometheus MBR | **17.7** | **26.2** | **3.35** |
| top-$p$ ($p = 0.9, t = 0.3$) | 14.3 | 23.7 | 0.40 |
| top-$p$ ($p = 0.9, t = 0.7$) | 14.7 | 23.9 | 0.70 |
| top-$p$ ($p = 0.5, t = 0.3$) | 15.3 | 24.9 | 1.50 |
| top-$p$ ($p = 0.5, t = 0.7$) | 15.0 | 24.9 | 1.35 |
| top-$k$ ($k = 50, t = 0.3$) | 14.7 | 23.6 | 0.55 |
| top-$k$ ($k = 50, t = 0.7$) | 15.0 | 23.5 | 0.65 |
| top-$k$ ($k = 20, t = 0.3$) | 15.0 | 24.7 | 1.25 |
| top-$k$ ($k = 20, t = 0.7$) | 14.7 | 24.0 | 0.75 |

Table 7: AlpacaEval 2.0 win rates for top-$p$ and top-$k$ decoding, along with the average score differences compared to greedy decoding across all models (denoted **Avg. $\Delta$**). Top-$p$ and top-$k$ sampling improve over greedy decoding, but do not match the performance improvements of Prometheus MBR decoding.

We find that top-$p$ and top-$k$ sampling improves over greedy decoding, and can achieve performance close to that of MBR decoding with ROUGE. The improvements are nonetheless much smaller than MBR decoding with Prometheus. Nonetheless, these results demonstrate that top-$p$ and top-$k$ sampling could be used to produce hypothesis sets containing higher quality candidates, which could in turn improve downstream MBR decoding performance.

## A.4 GENERATION LENGTHS

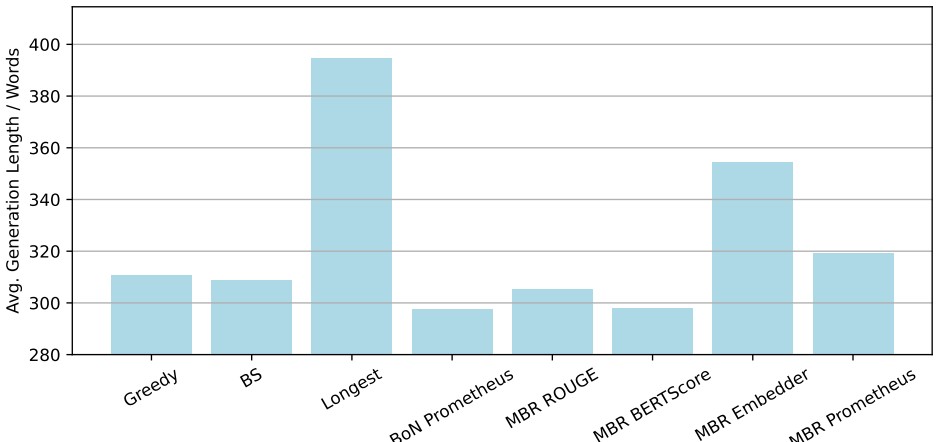

Figure 6: Average generation lengths (in words) for various decoding strategies, averaged over all five generator LLMs. MBR with Prometheus produces slightly longer outputs than most baseline methods, although these outputs are still far shorter than those produced by MBR with SFR-embedder and by the *longest decoding* baseline.

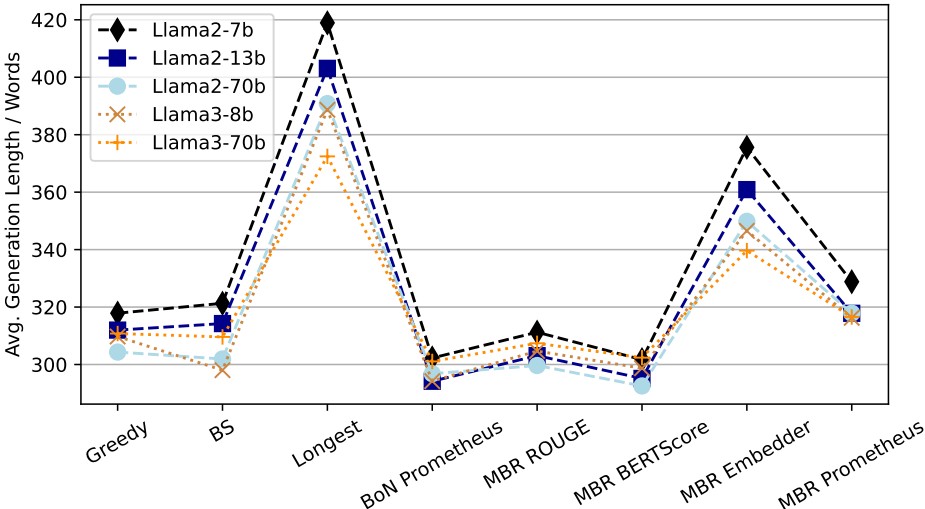

Figure 7: Average generation lengths (in words) for various models and decoding strategies on AlpacaEval. MBR with Prometheus produces slightly longer outputs than most baseline methods, although these outputs are still far shorter than those produced by MBR with SFR-embedder and by the *longest decoding* baseline.

We compute the average generation lengths (in words) of our five generator LLMs on AlpacaEval 2.0 with different decoding strategies and plot the results in Figures 6 and 7. We find that MBR decoding with Prometheus yields outputs that are on average longer than those yielded by greedy decoding, beam search or Prometheus BoN decoding. They are nonetheless much shorter than the outputs yielded by MBR decoding with SFR-Embedder and decoding by selecting the longest output. We hypothesise that MBR decoding with Prometheus encourages selection of more detailed responses which improves model performance, although we note that verbosity alone is not enough to improve performance on our benchmarks as the LONGEST baseline fails to yield any gains, and as we use length-controlled metrics for evaluation.

|  | Bullet List | Numbered List | Both | Neither |
|---|---|---|---|---|
| Greedy | 14.2 | 29.1 | 26.4 | 30.2 |
| Prometheus BoN | 13.6 | 29.8 | 26.0 | 30.8 |
| Prometheus MBR | 13.9 | 30.1 | 28.1 | 27.8 |

Table 8: Percentage of responses to AlpacaEval containing bullet lists, numbered lists, both bullet and numbered lists, and neither bullet nor numbered lists. These responses were generated by both Llama-3-8b-Instruct and Llama-3-70b-Instruct using various decoding strategies. We find that there is little difference in the formatting of MBR-generated, BoN-generated and greedy-decoded outputs.

|  | TTR | FK |
|---|---|---|
| Greedy | 0.514 | 12.13 |
| Prometheus BoN | 0.520 | 12.40 |
| Prometheus MBR | 0.521 | 12.24 |

Table 9: Averaged type token ratios (TTR) and Flesch Kincaid readability scores (FK) across outputs generated by both Llama-3-8b-Instruct and Llama-3-70b-Instruct using various decoding strategies. We find that there is little difference in lexical diversity (as measured by TTR) and readability (as measured by FK scores) between MBR-generated, BoN-generated and greedy-decoded outputs.

## A.5 Linguistic Features

We attempt to identify linguistic differences (beyond generation length) between MBR-generated responses and responses produced using greedy decoding and BoN decoding. We first consider looking for distinctive features in the formatting of MBR-generated responses by identifying responses to AlpacaEval that make use of bullet and numbered lists and then computing the percentage of outputs that fall into each category. We use Llama-3-8b-Instruct and Llama-3-70b-Instruct as the generator models and Prometheus-2-7b as the MBR judge, and document the results in Table 8. We find that there is little difference in the formatting of MBR-generated, BoN-generated and greedy-decoded outputs.

Next, we analyse the lexical diversity and readability of MBR-generated outputs, using the same dataset, generator and judge model as for the formatting experiments. For lexical diversity, we measure the type token ratio (Richards, 1987) of outputs, and for readability we compute Flesch Kincaid readability scores (Flesch, 1948). We document our findings in Table 9, and leave further analysis of linguistic differences between various decoding strategies to future work.

## A.6 Speculative Decoding

|  | Avg time per generation (s) | Avg tokens / s | GPT-4o score |
|---|---|---|---|
| Vanilla MBR | 59.3 | 4.48 | 6.83 |
| MBR + Spec. Decoding | 52.9 | 5.21 | 6.90 |

Table 10: Results demonstrating that MBR decoding can be combined with speculative decoding to increase autoregressive decoding speed. We measure the average time per generation and the average tokens per second associated MBR decoding with and without speculative decoding, and show that speculative decoding improves decoding speed without sacrificing output quality.

We can combine MBR decoding with speculative decoding (Leviathan et al., 2023) to speed up the MBR decoding step. We demonstrate this on 100 samples uniformly drawn from AlpacaEval, using Llama-3-70b-Instruct as both the generator and the MBR judge (see `huggingface.co/ibm-fms/Llama-3-70b-accelerator` for our chosen draft model), and report our findings in Table 10.

## A.7 Rank Correlation with GPT-4o Scores

We used GPT-4o as a reference-free judge to score the outputs of Llama-3-8b-Instruct ($N_{cand} = 30$, $t = 0.7$) on the first turn of MT-Bench. Outputs were scored on a scale of 1 to 10, using the standard MT-bench evaluation approach. We used a GPT-4o judge temperature of 0.5 and generated three scores per generation, taking as our final scores the average of the three sampled scores. Then, for

|  | Avg. Δ over Greedy | Avg. Rank Corr |
| --- | --- | --- |
| Prometheus-7b | 0.28 | 0.119 |
| Prometheus-8x7b | 0.39 | 0.136 |
| JudgeLM-7b | 0.22 | 0.053 |
| JudgeLM-33b | 0.31 | 0.113 |
| Llama-3-8b-Instruct | 0.28 | 0.116 |
| Llama-3-70b-Instruct | 0.41 | 0.144 |

Table 11: Spearman's rank correlation between the GPT-4o scores and MBR scores generated using various MBR judge models, along with average Δ over greedy decoding, computed on Llama-3-8b-Instruct outputs on turn 1 of MT-Bench. We find that stronger judges (as measured by the avg. Δ over greedy) are associated with higher average rank correlations with GPT-4o scores.

every sample, we computed the Spearman's rank correlation between the GPT-4o scores and the MBR scores.

We find that stronger judges (higher avg. Δ over greedy) are generally associated with better correlation with GPT-4o scores, although the absolute value of this correlation is not very high. This suggests that our judges are unlikely to be overly biased towards GPT-4o, although stronger judges will generally agree with GPT-4o more.

## B  GPT-4O-JUDGE DETAILS

We use GPT-4o as a judge for both AlpacaEval 2.0 and MT-Bench. More specifically, we use `gpt-4o-2024-05-13`, accessed through an Azure endpoint.

For AlpacaEval 2.0, we use the official AlpacaEval implementation[4] to conduct evaluation. We use the `weighted_alpaca_eval_gpt4_turbo` evaluator and baseline results against the default `gpt-4-1106-preview` generations unless other specified.

For MT-Bench, we use the single- and multi-turn judge prompts provided by LLMSYS FastChat[5] as our judge prompts. Due to stochasticity in the outputs of the judge models, even with temperature set to zero, we generate three judgements per sample and take as the final score the median of the three judgement scores.

## C  PROMETHEUS SCORING RUBRICS

Prometheus takes as input a scoring rubric that defines scoring criteria to be used during evaluation. We use a single, generic scoring rubric for all our experiments:

```
[Consider a wide range of factors such as the helpfulness,
relevance, accuracy, depth, creativity, and level of detail of
the response.]
Score 1:  The answer is completely unhelpful and incorrect.
Nothing useful can be learned from it.
Score 2:  The answer contains some helpful and useful information,
but major flaws, in terms of facuality, accuracy and relevance,
are also present.
Score 3:  The answer is mostly helpful and relevant, although
minor flaws exist.
Score 4:  The answer is accurate, relevant and helpful, although
there are some clear improvements that can be made with respect to
depth, creativity and detail.
Score 5:  The answer is excellent.  It is completely accurate and
relevant, and demonstrates a high degree of depth and creativity.
```

---

[4] github.com/tatsu-lab/alpaca_eval
[5] github.com/lm-sys/FastChat

We hypothesise that the performance of MBR and BoN decoding with Prometheus could be improved through further optimisation of the scoring rubric, particularly with question-specific adjustments, where unique rubrics are tailored to each question. We leave this to future work.

## D    ALPACAEVAL CATEGORIES

We classify questions from AlpacaEval and MT-Bench and then evaluate performance by category.

For AlpacaEval, we perform manual inspection on the dataset and identify ten common question categories. We then use GPT-4o to classify questions based on these categories, using the following prompt:

```
Categorise an instruction based on the following list of
categories.  Only choose one category, and only return the
category, nothing else.

- Creative writing
- Business, technical and scientific writing
- Argumentation, debate and persuasion
- Mathematical reasoning
- Puzzles and logical reasoning
- Coding
- How-to and other guides
- Recommendations and advice
- Factual question-answering
- Other

Example Instruction:  Bob has 5 sisters and 1 brother.  How many
siblings does one of Bob's sisters have?
Category:  Puzzles and logical reasoning
Example Instruction:  Write me a resignation email for my job as
an accountant, explaining that I am leaving to pursue my dream of
becoming a lion tamer.
Category:  Business, technical and scientific writing
Example Instruction:  What factors gave rise to the English Civil
War.  Category:  Factual question-answering
Example Instruction:  I am visiting Kyoto next April.  Recommend
me 10 things to do!
Category:  Recommendations and advice
Example Instruction:  Pretend to be Donald Trump - write a speech
announcing that you are becoming a Democrat.
Category:  Creative Writing
Instruction:  {instruction}
Category:
```

The percentage of questions assigned to them are listed in Table 12.

The results of our performance by category analysis for MBR decoding with Prometheus on AlpacaEval are illustrated in the main text, in Figure 3.

## E    LLAMA-3 AS AN MBR AND BON UTILITY METRIC

We experiment with using Llama-3-8b-Instruct and Llama-3-70b-Instruct as MBR and BoN utility metrics in Section 3.3.1. This entails prompting the models to act as reference-based or reference-free evaluators. Our prompt for single-turn reference-based evaluation is as follows:

| Category | Percentage of Questions |
|---|---|
| Factual question-answering | 24.3 |
| How-to and other guides | 20.6 |
| Recommendations and advice | 12.1 |
| Mathematical reasoning | 3.2 |
| Other | 2.2 |
| Creative writing | 11.8 |
| Business, technical and scientific writing | 12.7 |
| Puzzles and logical reasoning | 3.5 |
| Coding | 6.7 |
| Argumentation, debate and persuasion | 2.7 |

Table 12: AlpacaEval question categories identified by GPT-4o.

```
[Instruction]
Please act as an impartial judge and evaluate the quality of
the response provided by an AI assistant to the user question
displayed below.  In addition to the user question, you are
also given a reference answer.  This is the best possible answer
provided by a human expert.  You should evaluate the assistant's
response based on this.  A good assistant's answer should share
the content and style of the reference answer.  Begin your
evaluation by providing a short explanation.  Be as objective as
possible.  After providing your explanation, you must rate the
response on a scale of 1 to 10 by strictly following this format:
"[[rating]]", for example:  "Rating:  [[5]]".

[Question]
{question}

[Reference Answer]
{reference}

[The Start of Assistant's Answer]
{answer}
[The End of Assistant's Answer]
```

Our prompt for single-turn reference-free evaluation is as follows:

```
[Instruction]
Please act as an impartial judge and evaluate the quality of
the response provided by an AI assistant to the user question
displayed below.  Begin your evaluation by providing a short
explanation.  Be as objective as possible.  After providing your
explanation, you must rate the response on a scale of 1 to 10
by strictly following this format:  "[[rating]]", for example:
"Rating:  [[5]]".

[Question]
{question}

[The Start of Assistant's Answer]
{answer}
[The End of Assistant's Answer]
```

For multi-turn evaluation, we use a system prompt to specify the rules. For reference-based evaluation:

```
Please act as an impartial judge and evaluate the quality of
the response provided by an AI assistant to the user question
displayed below.  Your evaluation should focus on the assistant's
answer to the second user question.  In addition to the user
question and conversation history, you are also given a reference
answer.  This is the best possible answer to the second user
question provided by a human expert.  You should evaluate the
assistant's response based on this.  A good assistant's answer
should share the content and style of the reference answer.  Begin
your evaluation by providing a short explanation.  Be as objective
as possible.  After providing your explanation, you must rate the
response on a scale of 1 to 10 by strictly following this format:
"[[rating]]", for example:  "Rating:  [[5]]".
```

For multi-turn, reference-free evaluation:

```
Please act as an impartial judge and evaluate the quality of
the response provided by an AI assistant to the user question
displayed below.  Your evaluation should focus on the assistant's
answer to the second user question.  Begin your evaluation by
providing a short explanation.  Be as objective as possible.
After providing your explanation, you must rate the response on a
scale of 1 to 10 by strictly following this format:  "[[rating]]",
for example:  "Rating:  [[5]]".
```

The prompt template for both reference-based and reference-free multi-turn evaluation is:

```
<|The Start of Assistant A's Conversation with User|>

### User:
{question_1}

### Assistant A:
{answer_1}

### User:
{question_2}

### Assistant A:
{answer_2}

<|The End of Assistant A's Conversation with User|>
```

# F    REWARD MODEL AS A BON UTILITY METRIC

|  | 2-7B | 2-13B | 2-70B | 3-8B | 3-70B | Avg. $\Delta$ |
|---|---|---|---|---|---|---|
| Greedy | 5.72 | 5.90 | 6.50 | 7.54 | 8.29 | 0 |
| BS | 5.58 | 5.95 | 6.49 | 7.30 | 8.20 | -0.09 |
| Prometheus BoN | 5.77 | 6.08 | 6.65 | 7.66 | 8.42 | 0.13 |
| Starling-RM-7b BoN | 5.99 | **6.49** | **6.85** | **7.88** | 8.46 | **0.34** |
| Prometheus MBR | **6.10** | 6.26 | 6.79 | 7.69 | **8.50** | 0.28 |

Table 13: MT-Bench scores for various models and decoding strategies, along with the average score differences compared to greedy decoding across all models (denoted **Avg.** $\Delta$). BoN decoding with reward model StarlingRM-7b marginally outperforms MBR decoding with Prometheus.

We evaluate BoN decoding using Starling-RM-7B-alpha (Zhu et al., 2023a) as the utility metric. Starling-RM is a strong 7B sequence classifier reward model trained to facilitate RLHF (Stiennon et al., 2022) that achieves a similar overall score to Prometheus-2-7B on RewardBench (Lambert et al., 2024). It takes as inputs a prompt and a single candidate and outputs a scalar reward. As with reward models in general, Starling-RM does not support reference-based evaluation.

We find that BoN decoding with Starling-RM as a utility metric outperforms MBR decoding with Prometheus by a small margin. We have two possible explanations for this discrepancy. Firstly, Starling-RM achieves a much higher score than Prometheus-2-7B on RewardBench's Chat task (likely due to the distribution of its training data), suggesting that it may simply be more suited to our particular benchmarks. Secondly, by providing continuous scalar rewards instead of discrete integer scores, Starling-RM enables more fine-grained evaluation of candidate outputs, allowing it to distinguish between outputs that Prometheus might have rated equally. We believe that using a reference-based reward model as the metric for MBR decoding could combine the advantages of fine-grained scoring with the increased reliability of consensus-based output selection. We leave the development of such models to future work.

## G  SELF-TRAINING: ADDITIONAL RESULTS

### G.1  HEAD-TO-HEAD RESULTS

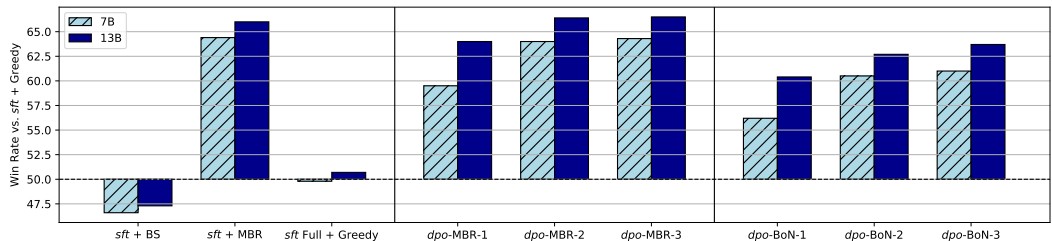

Figure 8: AlpacaEval 2.0 win rates (%) for self-trained models and various SFT baselines against *sft* with greedy decoding. Generation for all *dpo* models is done with greedy decoding. We find that MBR self-training with DPO allows models to match their MBR-decoding performance.

We conduct head-to-head evaluation of our DPO self-trained models and various SFT baselines against *sft* with greedy decoding using the AlpacaEval 2.0 and illustrate our findings in Figure 8. Our head-to-head results show that our MBR self-trained models outperform our BoN self-trained models, the *sft* with beam search baseline, and the full *sft* with greedy decoding baseline. Our MBR self-trained models match the performance of the *sft* with MBR decoding baseline.

### G.2  RESULTS ON NLP BENCHMARKS

|  | Model | MMLU (↑) | ARC challenge (↑) | HellaSwag (↑) | TruthfulQA (↑) |
|---|---|---|---|---|---|
| | *sft* | 47.2 | 57.3 | 80.6 | 51.6 |
| 7B | *dpo*-3-BoN | 47.2 | 57.5 | 80.6 | 53.5 |
| | *dpo*-3-MBR | 47.3 | 57.1 | 80.7 | 52.6 |
| | *sft* | 56.1 | 62.3 | 83.5 | 48.4 |
| 13B | *dpo*-3-BoN | 56.3 | 62.5 | 83.5 | 48.2 |
| | *dpo*-3-MBR | 56.1 | 62.6 | 83.5 | 47.4 |

Table 14: Evaluation results of Prometheus self-trained models on four different NLP benchmarks. We find that MBR and BoN self-training maintains performance on across all four datasets compared with the *sft* models.

We assess our self-trained models on four different NLP benchmarks: MMLU (Hendrycks et al., 2021), ARC challenge (Clark et al., 2018), HellaSwag (Zellers et al., 2019) and TruthfulQA (Lin et al., 2022), and report our results in Table 14. We find that self-training maintains performance across all four benchmarks despite us using a training dataset that is irrelevant for these tasks. This

shows that MBR self-training can be used to improve the instruction-following abilities of models without jeopardising other skills.

## G.3   MBR Self-Training with ROUGE

|  | AlpacaEval 2.0 | | MT-Bench | |
|---|---|---|---|---|
|  | **7B** | **13B** | **7B** | **13B** |
| *sft* | 5.18 | 8.24 | 5.43 | 5.85 |
| *dpo*-1-MBR-Prometheus | 5.68 | 10.8 | 5.78 | 6.48 |
| *dpo*-2-MBR-Prometheus | 7.22 | 13.9 | 6.11 | 6.73 |
| *dpo*-3-MBR-Prometheus | **8.86** | **15.3** | **6.14** | **6.75** |
| *dpo*-1-MBR-ROUGE | 4.66 | 5.61 | 7.65 | 5.98 |
| *dpo*-2-MBR-ROUGE | 5.83 | 5.78 | 9.01 | 6.06 |
| *dpo*-3-MBR-ROUGE | 5.42 | 5.67 | 8.31 | 5.91 |

Table 15: AlpacaEval 2.0 win rates (%) and MT-Bench scores for models self-trained using DPO with Prometheus and with ROUGE-1 as utility metrics. MBR self-training with ROUGE fails to yield substantial gains.

We explore using ROUGE-1 as the utility metric for MBR self-training. We find that MBR self-training with ROUGE fails to yield substantial gains. Improvements also saturate quickly, with model performance decreasing after the third training iteration. These results highlight the importance of choosing the correct MBR utility metric for self-training.

## G.4   MBR Self-Training with Llama-3-8b-Instruct

We also consider using Llama-3-8b-Instruct in place of Prometheus as the utility metric to self-train Llama-3-8b. We compare this approach to using Prometheus to train Llama-3-8b, and find that both approaches lead to significant gains. The fact that Llama-3-8b-Instruct as the judge can be used to improve Llama-3-8b suggests that self-improvement using MBR decoding is possible. We leave this to future work.

|  | **Prometheus** | **Llama-3-8b-Instruct** |
|---|---|---|
| *sft* | 6.70 | 6.70 |
| *dpo*-1-MBR | 6.94 | 6.99 |
| *dpo*-2-MBR | 7.45 | 7.51 |
| *dpo*-3-MBR | 7.55 | 7.52 |

Table 16: MT-Bench scores for Llama-3-8b self-trained using DPO with either Prometheus or Llama-3-8b-Instruct as the MBR judge. Generation is done using greedy decoding for all models. We find that both approaches lead to significant gains.

## G.5   Alternative Pair Selection Strategies

|  | AlpacaEval 2.0 | MT-Bench |
|---|---|---|
| *sft* | 5.18 | 5.43 |
| *dpo*-1-MBR$_{BW}$ | 5.68 | 5.78 |
| *dpo*-2-MBR$_{BW}$ | 7.22 | 6.11 |
| *dpo*-3-MBR$_{BW}$ | 8.86 | **6.14** |
| *dpo*-1-MBR$_{BMW}$ | 4.95 | 5.78 |
| *dpo*-2-MBR$_{BMW}$ | 8.06 | 5.96 |
| *dpo*-3-MBR$_{BMW}$ | **8.88** | 5.97 |

Table 17: AlpacaEval 2.0 win rates (%) and MT-Bench scores for models self-trained using DPO with BW and BMW preference pair selection strategies with *sft*-7b as the base model. Generation is done using greedy decoding for all models. We find that our BMW models perform similarly to the original BW models on AlpacaEval 2.0 and slightly worse on MT-Bench.

We investigate using an alternative MBR preference pair selection strategy. Following Yang et al. (2024), we create two preference pairs from each sample, one formed from the highest-scoring (best) and median-scoring (mid) outputs, and another formed from the median-scoring and lowest-scoring

(worst) outputs. We follow the exact same DPO training procedure as before, but replace our original BW training set with this new BMW training set.

We document our results in Table 17. We find that our BMW models perform similarly to the original BW models on AlpacaEval 2.0 and slightly worse on MT-Bench. We do not pursue this line of work further and leave additional investigations to future work.

### G.6 PERFORMANCE BY CATEGORY

| Category | Percentage of Questions |
|---|---|
| Factual question-answering | 26.4 |
| How-to and other guides | 21.1 |
| Recommendations and advice | 4.4 |
| Mathematical reasoning | 0.1 |
| Other | 1.3 |
| Creative writing | 18.6 |
| Business, technical and scientific writing | 19.0 |
| Puzzles and logical reasoning | 0.1 |
| Coding | 6.3 |
| Argumentation, debate and persuasion | 2.7 |

Table 18: UltraChat-200k question categories identified by GPT-4o. We perform this analysis on a subsample of 1000 prompts sampled randomly from our 12000 training prompts.

We replicate the question category analysis described in Section 3.3 and Appendix D for our Prometheus MBR self-trained models and report results in Figure 4. We find that performance relative to the *sft* models improves across almost all question categories, with performance on writing-based categories improving more than on reasoning-based categories. Performance on mathematical reasoning remains unchanged for the 7B model and decreases for the 13B model.

To better understand this discrepancy, we sample 1000 prompts from our 12000 UltraChat training prompts and categorise them using GPT-4o, following the same procedure described in Section 3.3 and Appendix D. We document our results in Table 18. We find that reasoning-based questions (coding, puzzles and logical reasoning, mathematical reasoning) are underrepresented in this dataset, with mathematical reasoning and puzzles and logical reasoning especially underrepresented. We attribute our models' inconsistent improvements in these areas to this lack of data.

### G.7 GENERATION LENGTHS

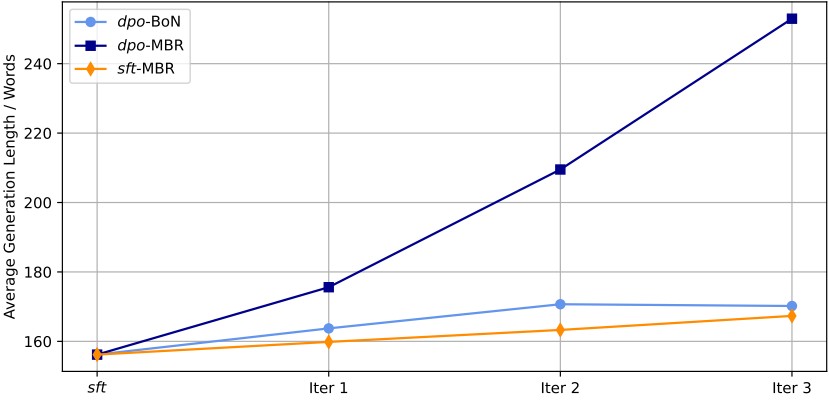

Figure 9: Average generation lengths (in words) after iterative Prometheus MBR and BoN self-training on AlpacaEval. MBR self-training with DPO teaches the model to generate more detailed responses.

We measure the generation lengths of our Prometheus MBR and BoN self-trained models on AlpacaEval. We find that MBR self-training with DPO teaches the model to generate longer and more detailed responses at every iteration. In contrast, BoN self-training and MBR self-training with SFT only results in small increases in generation lengths.

# H HUMAN STUDY

|  | Win | Draw | Loss |
|---|---|---|---|
| Prometheus MBR vs. Greedy | 30.0 | 53.3 | 16.7 |
| Prometheus BoN vs. Greedy | 21.7 | 60.0 | 18.3 |

Table 19: Head-to-head evaluation of Prometheus MBR and Prometheus BoN vs. greedy decoding for Llama-2-70b conducted on the AlpacaEval dataset by human evaluators.

|  | Win | Draw | Loss |
|---|---|---|---|
| *dpo*-3-MBR vs. *sft* | 46.7 | 43.3 | 10.0 |
| *dpo*-3-BoN vs. *sft* | 33.3 | 55.0 | 11.7 |

Table 20: Head-to-head evaluation of 13B *dpo*-3-MBR and *dpo*-3-BoN vs. *sft* conducted on the AlpacaEval dataset by human evaluators. Greedy decoding used for all models.

We conduct a human study for key MBR inference (Section 3) and MBR distillation (Section 4) experiments. The objective of this study is to verify that our LLM evaluation results (AlpacaEval 2.0 and MT-Bench) align with real human judgements.

We recruited 4 volunteers, each of whom were given 60 samples in total to evaluate. Each sample consisted of a randomly-sampled AlpacaEval prompt and two corresponding generations displayed in a random order. Volunteers were asked to select their preferred generation and, if neither generation was preferred, to then rate the generations as equal.

We conducted four head-to-head experiments. The first two experiments, corresponding to experiments in Section 3, were between Prometheus MBR vs. greedy decoding and Prometheus BoN vs. greedy decoding with Llama-2-70b. The next two experiments, corresponding to experiments in Section 4, were between 13B *dpo*-3-MBR vs. *sft* and *dpo*-3-BoN vs. *sft*. We used 60 samples for each experiment. Volunteers were given an even distribution of samples from each experiment.

Our results show good alignment with our LLM evaluation results. We find that Prometheus MBR decoding performs well against greedy decoding, with its win rate higher than the corresponding win rate for Prometheus BoN decoding vs. greedy decoding. We also find that *dpo*-3-MBR significantly outperforms *sft*, and its margin of victory is greater than the margin of victory of *dpo*-3-BoN vs. *sft*. Furthermore, the margin of improvement associated with MBR distillation is larger than the corresponding margin for MBR inference. These findings align with our findings from our automatic LLM evaluation experiments.

# I ALGORITHMS

## I.1 MBR INFERENCE

We provide the algorithm for MBR inference in Algorithm 1, complementing the mathematical overview provided in Section 2.

## I.2 MBR DISTILLATION

In MBR distillation, we first gather a preference dataset using MBR decoding over a set of input prompts. Given input prompts $X_k$ and a base model $\pi_{\theta_{k-1}}$, we first sample $N_{\text{cand}}$ outputs per prompt $y^{(i)} \sim \pi_{\theta_{k-1}}(\cdot|x)$, where $x \in X_k$, forming hypothesis set $\mathcal{H}_{\text{hyp}} = \{y^{(1)}, y^{(2)}, \ldots, y^{(N_{\text{cand}})}\}$.

We then compute expected utility for elements in $\mathcal{H}_{\text{hyp}}$

$$\tilde{u}(y^{(i)}) = \frac{1}{N_{\text{cand}}} \sum_{j=1}^{N_{\text{cand}}} u(y^{(i)}, y^{(j)}) \tag{4}$$

---

**Algorithm 1** MBR Inference

---

**Inputs:** Prompt $x$, generator model $p$, reference-based utility metric $u$, number of candidates $N_{\text{cand}}$, sampling temperature $t$.
**Output:** MBR output $\hat{y}$

   Initialise $\mathcal{H}_{\text{hyp}} \leftarrow \varnothing$
   **for** $i \in \{1, 2, \ldots, N_{\text{cand}}\}$ **do**
     *Sample* $y^{(i)} \sim p(\cdot|x)$ with temperature $t$
     *Add* $y^{(i)}$ to $\mathcal{H}_{\text{hyp}}$                           // Form hypothesis set
   **end for**
   **for** $y^{(i)} \in \mathcal{H}_{\text{hyp}}$ **do**
     *Compute* $\tilde{u}(y^{(i)}) = \frac{1}{N_{\text{cand}}} \sum_{j=1}^{N_{\text{cand}}} u(y^{(i)}, y^{(j)})$              // Compute expected utility
   **end for**
   *Select* $\hat{y} = \arg\max_{y \in \mathcal{H}_{\text{hyp}}} \tilde{u}(y)$
   **return** $\hat{y}$

---

and form preference pairs using the output that maximises expected utility $\hat{y}^+$ and the output that minimises it $\hat{y}^-$.

$$\hat{y}^+ = \arg\max_{y \in \mathcal{H}_{\text{hyp}}} \tilde{u}(y)$$

$$\hat{y}^- = \arg\min_{y \in \mathcal{H}_{\text{hyp}}} \tilde{u}(y)$$

We collect these preference pairs and their prompt $(x, \hat{y}^+, \hat{y}^-)$ in a dataset we denote $\mathcal{Y}_k$.

We then use these preference pairs for DPO (Rafailov et al., 2024) training. In DPO, we minimise the following policy objective:

$$\mathcal{L}_{\text{DPO}} = -\mathbb{E}_{(x, \hat{y}^+, \hat{y}^- \sim \mathcal{Y}_k)} \left[ \log \sigma \left( \beta \log \frac{\pi_\theta(\hat{y}^+|x)}{\pi_{\text{ref}}(\hat{y}^+|x)} - \beta \log \frac{\pi_\theta(\hat{y}^-|x)}{\pi_{\text{ref}}(\hat{y}^-|x)} \right) \right] \tag{5}$$

where $\pi_\theta$ is the policy and $\pi_{\text{ref}}$ the reference model. We repeat this process iteratively with $k = 1, 2, \ldots, K$. The initial model $\pi_{\theta_0}$ is the base *sft* model. We choose $K = 3$ in our experiments.

We also provide the algorithm for MBR distillation with DPO in Algorithm 2.

## J LIMITATIONS

While our work demonstrates the significant potential of MBR decoding, there are limitations that should be addressed in future research. Firstly, although we demonstrate using existing judge LLMs utility metrics that MBR decoding consistently outperforms BoN decoding, this does not preclude the existence of reference-free metrics that *are* powerful enough to match or surpass the performance of their direct reference-based counterparts. This relates to a possible broader limitation on the benefits of using consensus quality for output selection, as the consensus solution may not always be the optimal one. We encourage future work to train better utility metrics in order to better understand these limitations. A second limitation of our work is that we do not study the biases introduced by the utility metric. One particularly pernicious form of bias is "reward-hacking" behavior, where the utility metric (likely as a result of its own training) selects outputs that evaluate well on our benchmarks but that are actually worse in quality. While we preclude this from being the case in our experiments via our human study (Appendix H), this does not mean that such pernicious behavior cannot arise in other settings. Finally, we do not study limitations on scalability. Although we show that small judge LLMs (7B) can serve as utility metrics for much larger models (70B), it is likely that weaker utility metrics cease being useful for very strong LLMs and on very complex tasks. Further research is needed to determine when this breakdown occurs.

---

**Algorithm 2** MBR Distillation with DPO

---

**Inputs:** Prompt sets $X_1, X_2, \ldots, X_K$, *sft* model $\pi_{\theta_0}$, reference-based utility metric $u$, number of candidates $N_{\text{cand}}$, sampling temperature $t$, number of self-training iterations $K$.
**Output:** Self-trained model $\pi_{\theta_K}$

$\quad$ **for** $k \in \{1, 2, \ldots, K\}$ **do**
$\qquad$ Initialise $\mathcal{Y}_k \leftarrow \varnothing$
$\qquad$ **for** $x \in X_k$ **do**
$\qquad\quad$ Initialise $\mathcal{H}_{\text{hyp}} \leftarrow \varnothing$
$\qquad\quad$ **for** $i \in \{1, 2, \ldots, N_{\text{cand}}\}$ **do**
$\qquad\qquad$ *Sample* $y^{(i)} \sim \pi_{\theta_{k-1}}(\cdot|x)$ with temperature $t$
$\qquad\qquad$ *Add* $y^{(i)}$ to $\mathcal{H}_{\text{hyp}}$ $\qquad\qquad\qquad\qquad\qquad\qquad\qquad$ // Form hypothesis set
$\qquad\quad$ **end for**
$\qquad\quad$ **for** $y^{(i)} \in \mathcal{H}_{\text{hyp}}$ **do**
$\qquad\qquad$ *Compute* $\tilde{u}(y^{(i)}) = \frac{1}{N_{\text{cand}}} \sum_{j=1}^{N_{\text{cand}}} u(y^{(i)}, y^{(j)})$ $\qquad$ // Compute expected utility
$\qquad\quad$ **end for**
$\qquad\quad$ *Select* $\hat{y}^+ = \arg\max_{y \in \mathcal{H}_{\text{hyp}}} \tilde{u}(y)$ $\qquad\qquad\qquad$ // Select highest scoring output
$\qquad\quad$ *Select* $\hat{y}^- = \arg\min_{y \in \mathcal{H}_{\text{hyp}}} \tilde{u}(y)$ $\qquad\qquad\qquad$ // Select lowest scoring output
$\qquad\quad$ *Add* $(\hat{y}^+, \hat{y}^-)$ to $\mathcal{Y}_k$ $\qquad\qquad\qquad\qquad\qquad$ // Form preference pairs
$\qquad$ **end for**
$\qquad$ *Update* $\pi_{\theta_k} \leftarrow DPO(\pi_{\theta_{k-1}}, \mathcal{Y}_k)$ $\qquad\qquad$ // DPO training on preference pairs
$\quad$ **end for**
$\quad$ **return** $\pi_{\theta_K}$

## K $\quad$ TRAINING AND INFERENCE HYPERPARAMETERS

| Hyperparameter | Value |
|---|---|
| Learning Rate | 5$e$-6 |
| Num Epochs | 3 |
| Batch Size | 32 |
| Optimiser | AdamW |
| $\beta_1$ | 0.9 |
| $\beta_2$ | 0.95 |
| $\epsilon$ | 1$e$-8 |
| Weight Decay | 0.1 |
| Scheduler | Cosine |

Table 21: Hyperparameters for SFT and MBR self-training with SFT.

| Hyperparameter | Value |
|---|---|
| Learning Rate | 5$e$-7 |
| Num Epochs | 5 |
| Batch Size | 8 |
| Optimiser | RMSProp |
| $\alpha$ | 0.99 |
| $\beta_{\text{DPO}}$ | 0.1 |
| Scheduler | Constant with warmup |
| Warmup Steps | 150 |

Table 22: Hyperparameters for MBR self-training with DPO.

Our SFT and DPO hyperparameters for our self-training experiments in Section 4.2 are provided in Tables 21 and 22. We use `bf16` mixed precision training with 8xA100 GPUs for all experiments.

For inference, we use 4xA100 GPUs with `bf16` quantisation for all LLMs and judge LLMs, other than for the *Analysis of compute costs* experiments in Section 4.2, where we use 2xA100 GPUs. We use vLLM (Kwon et al., 2023) as the inference engine for all experiments.

