# OpenReview forum: "Better Instruction-Following Through Minimum Bayes Risk"
_ICLR.cc/2025/Conference — ICLR 2025 Spotlight_

### Official Review · Reviewer_Z8nY · 2024-10-25

**Soundness:** 3
**Presentation:** 4
**Contribution:** 3
**Rating:** 8
**Confidence:** 3

**Summary:**

This paper explores MBR decoding with reference-based LLM judges for selecting one of n outputs for instruction finetuned LLMs at inference time.
The paper uses Prometheus2, LLama3, and JudgeLM models as a judge for Llama 2 & 3 models and evaluates on AlpacaEval and MT-Bench, comparing MBR decoding with best-of-n inference, and other one-best decoding strategies, and explore non-LLM utility functions for MBR as well.
They find gains across the bench, with smaller judges also being successful as utility function in MBR decoding for guiding larger LLMs. MBR decoding is furthermore combined with self-training (DPO) to yield further gains and overcome the added decoding costs.

**Strengths:**

1. The experiments are thorough (and well documented), including multiple sizes of learner and judge models, and multiple utility functions for comparison.
2. The analyses are interesting and well presented, explained and visualized. Important ablations, such as hypotheses size and temperature, are included.
3. The distillation solution is very attractive for boosting performance in practice without sacrificing inference time, which is contributing to a timely trend. There's a good chance this approach will be added to everyone's box of inference tricks.
4. Compute costs are well discussed and addressed with the distillation solution.
Solid and interesting review of past works, especially about Machine Translation where MBR decoding has already become popular.

**Weaknesses:**

1. The method is not particularly novel, as MBR decoding with LLMs has been studied in the past (as discussed in Related Work), but not with an LLM as judge.
2. Given the novelty being the use of a judge to find the consensus output, I would love to see more analysis on the dependence of the LLM judge’s quality. There is a notable gap with other utility functions, so it would be helpful to understand where they diverge. See questions below.

**Questions:**

1. Could the LLM model itself be used as a judge? In the spirit of self-improvement.
Jinnai et al. evaluated their utility metrics on benchmarks without LLM judge. How would LLM judges perform there? (e.g. machine translation, summarization)
2. Could there be some kind of overfitting/bias towards the GPT-4o judge that is dominating the win-rates in the comparison (the “closer” the judge to GPT-4o, the better the MBR)? In the extreme case, what if GPT-4o was used as a judge - I’m sure that would beat even Prometheus. Perhaps one could compare the outputs selected by the evaluated judges to GPT-4o decisions to find out if the ranking of win-rates corresponds to agreeing with GPT-4o.
3. I would love to see some qualitative analysis on how BoN and MBR generation selection differ with the same judge. The paper explains that the smoothing effect of MBR might be helpful to find the best generation, but I would like this to be made a little more built out and supported by examples.

---

> ### Author Response · Authors · 2024-11-24
> **Response to Reviewer Z8nY (1/2)**
>
> Thank you for your review! We are glad to hear you find the experiments thorough, the analysis interesting, and the distillation solution timely. We respond to your concerns in the response below, and we would be happy to engage in further discussion!
>
> ---
> *Question 1: Could the LLM model itself be used as a judge? In the spirit of self-improvement. Jinnai et al. evaluated their utility metrics on benchmarks without LLM judge. How would LLM judges perform there? (e.g. machine translation, summarization)*
>
> ---
>
> **Could the LLM model itself be used as a judge? In the spirit of self-improvement.**
>
> This is a great question. There are definitely strong hints that a strong LLM could be used as the judge both for MBR inference and distillation.
>
> In Section 3.3.1, we use Llama-3-70b-Instruct as both the judge and the generator LLM, and find (Table 3) that both MBR decoding and BoN decoding yield gains over greedy decoding (8.29 -> 8.35 for BoN and 8.29 -> 8.52 on MT-Bench for BoN and MBR respectively).
>
> Additionally (new in the updated manuscript), we use Llama-3-8b-Instruct as both the judge and the generator LLM (see Table 3 in the updated paper), and find that both MBR decoding and BoN decoding yield gains over greedy decoding (7.54 -> 7.60 for BoN and 7.54 -> 7.80 on MT-Bench for BoN and MBR respectively).
>
> Both results demonstrate that the generator and judge LLM can be the same LLM, and that MBR decoding is still better than BoN decoding in this setup.
>
> We also consider using Llama-3-8b-Instruct in place of Prometheus as the utility metric to self-train Llama-3-8b. We compare this approach to using Prometheus to train Llama-3-8b.
>
> | Method             | Prometheus | Llama-3-8b-Instruct |
> |--------------------|------------|---------------------|
> | STF                | 6.70       | 6.70                |
> | MBR DPO-1          | 6.94       | 6.99                |
> | MBR DPO-2          | 7.45       | 7.51                |
> | MBR DPO-3          | 7.55       | 7.52                |
>
> We see that generator LLMs can be used to improve LLMs. These distillation results have been added to Appendix G.7 in the updated manuscript.
>
> ---
>
> **Jinnai et al. evaluated their utility metrics on benchmarks without LLM judge. How would LLM judges perform there? (e.g. machine translation, summarization)**
>
> We ran additional experiments where we evaluated MBR with Prometheus along with greedy decoding and MBR with a task-specific metric for XSUM (summarisation) and WMT-19 Cs-En (translation), using Llama-3-8b-Instruct as the generator model.
>
> *XSUM (evaluated with BERTScore)*
>
> | Method                | Score   |
> |-----------------------|---------|
> | Greedy                | 69.45   |
> | MBR with ROUGE-L      | 69.72   |
> | MBR with Prometheus   | 69.24   |
>
> *WMT (evaluated with COMET)*
>
> | Method                | Score   |
> |-----------------------|---------|
> | Greedy                | 84.40   |
> | MBR with BLEURT       | 84.60   |
> | MBR with Prometheus   | 84.37   |
>
> We find that MBR with Prometheus does not outperform greedy decoding or MBR decoding with task specific metrics. We suspect that our task-specific metrics (ROUGE-L and BLEURT) are better suited to the closed-form tasks of (extreme) summarisation and translation than Prometheus. For translation, the lack of multilingual capabilities for the Prometheus model might also play a role (Prometheus was only trained on English finetuning data). Nonetheless we highlight that the differences in performance between these methods is small.

---

> ### Author Response · Authors · 2024-11-24
> **Response to Reviewer Z8nY (2/2)**
>
> *Question 2: Could there be some kind of overfitting/bias towards the GPT-4o judge that is dominating the win-rates in the comparison (the “closer” the judge to GPT-4o, the better the MBR)? In the extreme case, what if GPT-4o was used as a judge - I’m sure that would beat even Prometheus. Perhaps one could compare the outputs selected by the evaluated judges to GPT-4o decisions to find out if the ranking of win-rates corresponds to agreeing with GPT-4o.*
>
> ---
>
> **Could there be some kind of overfitting/bias towards the GPT-4o judge that is dominating the win-rates in the comparison (the “closer” the judge to GPT-4o, the better the MBR)?**
>
> This is a good point and definitely a reasonable concern.
>
> To ameliorate this concern, we conducted a human study (Appendix H, included in both the original and updated manuscript) to rule out the possibility of win-rates increasing solely due to overfitting to the GPT-4o judge. From our human study, we find that humans generally prefer MBR decoded outputs more than BoN or greedy outputs, suggesting that the increase in win-rates is due to true quality improvements.
>
> We also tried searching for distinctive linguistic characteristics in the MBR-decoded outputs relative to the greedy outputs, and documented our findings in our response to Reviewer Nmeb. We find none of the common characteristics indicative of reward hacking behavior (e.g. significantly increased verbosity, use of certain formatting tricks), which further suggests that MBR decoding with Prometheus does not overfit to a particular quirk of the GPT-4o judge and is instead associated with genuine quality improvements.
>
> Please see our response to Reviewer hFLa for further discussion on related ideas.
>
> ---
>
> **Perhaps one could compare the outputs selected by the evaluated judges to GPT-4o decisions to find out if the ranking of win-rates corresponds to agreeing with GPT-4o.**
>
> This is a great idea.
>
> We used GPT-4o as a reference-free judge to score the outputs of Llama-3-8b-Instruct (N_cand = 30, t = 0.7) on the first turn of MT-Bench. Outputs were scored on a scale of 1 - 10. We used a GPT-4o judge temperature of 0.5 and generated three scores per generation, taking as our final scores the average of the three sampled scores. Then, for every sample, we computed the Spearman’s rank correlation between the GPT-4o scores and the MBR scores:
>
> | Method                  | Avg. Delta over Greedy | Avg. Corr |
> |-------------------------|-----------------------|-----------|
> | Prometheus-7b           | 0.28                  | 0.119     |
> | Prometheus-8x7b         | 0.39                  | 0.136     |
> | JudgeLM-7b              | 0.22                  | 0.053     |
> | JudgeLM-33b             | 0.31                  | 0.113     |
> | Llama-3-8b-Instruct     | 0.28                  | 0.116     |
> | Llama-3-70b-Instruct    | 0.41                  | 0.144     |
>
> We find that stronger judges (higher Avg. Delta over Greedy) are generally associated with slightly better correlation with GPT-4o scores, although the absolute value of this correlation is not very high. This suggests that our judges are unlikely to be overly biased towards GPT-4o, although stronger judges will generally agree with GPT-4o more.
>
> We will add these findings to our camera-ready paper.
>
> ---
>
> *Question 3: I would love to see some qualitative analysis on how BoN and MBR generation selection differ with the same judge. The paper explains that the smoothing effect of MBR might be helpful to find the best generation, but I would like this to be made a little more built out and supported by examples.*
>
> We agree that this would be very beneficial! Please see our response to Reviewer Nmeb for an overview of our attempts to search for obvious linguistic characteristics that set BoN and MBR decoded outputs apart.

---

> ### Author Response · Authors · 2024-11-29
> **We eagerly await your response**
>
> Dear Reviewer Z8nY,
>
> We have responded to your review of our work and have updated our manuscript. We greatly appreciate the time you have taken to help us improve our paper!
>
> With the author-discussion period drawing to a close, we would be grateful if you could respond to our rebuttal - your feedback is crucial to the progress of our work and we would like our paper to be in the best possible shape before the discussion period closes.

---

### Official Review · Reviewer_hFLa · 2024-11-04

**Soundness:** 3
**Presentation:** 1
**Contribution:** 3
**Rating:** 6
**Confidence:** 4

**Summary:**

The paper presents a novel approach to improving the test-time performance of instruction-following LLMs through the application of Minimum Bayes Risk (MBR) decoding. The authors leverage LLM judges as reference-based evaluators to select high-quality outputs from a set of candidate outputs. They demonstrate that MBR decoding with LLM judges significantly outperforms greedy decoding and other decoding methods without references on benchmarks like AlpacaEval and MT-Bench. Furthermore, the paper explores iterative self-training on MBR-decoded outputs to retain performance improvements while mitigating additional test-time costs. The authors find that self-training using Direct Preference Optimisation leads to significant performance gains, matching or exceeding the performance of base models with MBR decoding.

**Strengths:**

1. The application of MBR decoding with LLM judges is a creative approach that combines recent advances in LLM evaluation with decoding techniques.
2. The experiments are thorough, and the benchmarks used are relevant and well-established in the field.
3.  This paper also explores the guiding role of MBR decoding in LLM’s DPO training, which is inspiring in subsequent model training.

**Weaknesses:**

Main weakness:
- The presentation lacks clarity and reads more like an experimental report than an academic paper. The methods section is merged with the experiments and results, lacking any formal formulations. For instance, in Section 4.1.2, Iterative DPO on MBR-Decoded Outputs, adding mathematical formulations would improve both understanding and reproducibility of the approach.

Others:
- This paper only compares some relatively simple decoding methods. If some better decoding methods such as speculative decoding and medusa can be added, the method will be more credible.
- The paper could also benefit from a discussion on the computational costs associated with MBR decoding and self-training, especially when scaling to larger models or datasets.

**Questions:**

- **Question 1:** How does the performance of MBR decoding with LLM judges compare to other state-of-the-art decoding methods beyond those presented in the paper?
- **Question 2:** Can the authors elaborate on any potential negative impacts of using LLM judges, such as the risk of overfitting to the judge's biases?
- **Question 3:** The hyperparameter N~cond~ is set to 12 for generating candidates for DPO but increased to 30 during decoding. How did the authors determine these values, and how do they align with the experiment illustrated in Figure 2?

---

> ### Author Response · Authors · 2024-11-24
> **Response to Reviewer hFLa (1/2)**
>
> Thank you for your review! We’re glad to hear you find our approach creative, our experiments thorough, and our benchmarks well-selective. We respond to your concerns in the response below, and we would be happy to engage in further discussion!
>
> ---
> *Main weakness: The presentation lacks clarity and reads more like an experimental report than an academic paper. The methods section is merged with the experiments and results, lacking any formal formulations. For instance, in Section 4.1.2, Iterative DPO on MBR-Decoded Outputs, adding mathematical formulations would improve both understanding and reproducibility of the approach.*
>
> To improve understanding and reproducibility of MBR distillation (iterative DPO), we have now included a mathematical formulation for it in Appendix I.2 and have referenced this in Section 4.1.2 - please see our updated manuscript. We hope this provides sufficient clarity regarding our distillation method.
>
> As our main contribution is exploring the use of LLM as a judge for MBR decoding, we provide a formal, mathematical description of MBR decoding in the Background section (Section 2) rather than in the experimental sections (Sections 3 and 4). We hope this provides sufficient clarity regarding MBR Inference.
>
> To add further clarity on both MBR Inference and MBR Distillation, we have also added to Appendix I detailed algorithm descriptions of both methods. Please let us know if there is anything more specific you’d like to see added!
>
> On the structuring of our paper: as our work is composed of two experimental sections - MBR Inference (Section 3) and MBR Distillation (Section 4) - it is difficult for us to maintain entirely separate methods and results sections. Within each experimental section however we have kept methods (3.1, 4.1), experimental results (3.2, 4.2) and further experiments (3.3) separate. We hope this is still able to provide a good level of clarity. Please let us know if you have any suggested formatting changes!
>
> ---
>
> *Other weakness: This paper only compares some relatively simple decoding methods. If some better decoding methods such as speculative decoding and medusa can be added, the method will be more credible.*
>
> Thank you for bringing up acceleration methods such as speculative decoding - this is a very interesting idea that we did not originally consider looking at in the context of MBR decoding.
>
> Methods like speculative decoding and Medusa are used to increase autoregressive decoding speed without impacting output quality, and therefore can be used alongside MBR decoding, which selects over decoded sequences to improve the final output quality.
>
> To demonstrate this, we applied speculative decoding to the LLM judge decoding step and measured changes to decoding speed. We used Llama-3-70b-Instruct as the MBR LLM judge, using ibm-fms/llama3-70b-accelerator (https://huggingface.co/ibm-fms/llama3-70b-accelerator) as the draft model and 100 prompts randomly sampled from AlpacaEval as the dataset. We used Llama-2-7b-chat as the generator model, without speculative decoding. All inference was done using 8xA100 GPUs.
>
> | Method                  | Avg time per generation (s) | Tokens / s | GPT-4 score |
> |-------------------------|-----------------------------|------------|-------------|
> | Vanilla MBR             | 59.3                        | 4.48       | 6.83        |
> | MBR + Spec. Decoding    | 52.9                        | 5.21       | 6.90        |
>
> From our results above, we see that speculative decoding can be used to improve MBR decoding speed with no loss in performance. We note however that MBR decoding is typically compute bound (as we do batch inference during decoding), so the speed increase is smaller than what you might see in the memory bound case (small batch sizes, where speculative decoding is most useful). We will incorporate discussion of these results in the camera-ready paper.
>
> ---
>
> *Other weakness: The paper could also benefit from a discussion on the computational costs associated with MBR decoding and self-training, especially when scaling to larger models or datasets.*
>
> We discuss the computational costs associated with MBR decoding and self-training in Section 4.2, where we demonstrate experimentally that (1) MBR decoding at inference time incurs a significant cost, largely associated with the decoding (utility metric calculation) step and (2) self-training mitigates this cost entirely, enabling the model to achieve greedy-decoding levels of throughput (tokens / s) with MBR decoding levels of performance. Please let us know if there is anything further you’d like us to discuss.

---

> ### Author Response · Authors · 2024-11-24
> **Response to Reviewer hFLa (2/2)**
>
> *Question 1: How does the performance of MBR decoding with LLM judges compare to other state-of-the-art decoding methods beyond those presented in the paper?*
>
> We believe that we have already selected state-of-the-art decoding methods to compare MBR decoding against, including BoN decoding and Universal Self-Consistency (Appendix A.2). If there are any other specific decoding methods we may have missed that you would like us to compare against, please let us know!
>
> ---
>
> *Question 2: Can the authors elaborate on any potential negative impacts of using LLM judges, such as the risk of overfitting to the judge's biases?*
>
> This is a very good point and definitely a possible concern! By using a judge LLM to select outputs, we inherently bias our outputs towards the preferences of the judge LLM. If these preferences are not actually associated with improved quality, then MBR decoding could have a negative overall impact. This is especially true for MBR distillation, as we repeatedly distill these outputs back into the model! We include a separate discussion of this idea in a new Limitations section, which we have added to Appendix J in the updated manuscript.
>
> However, we do not believe that overfitting to the judge LLMs occurs in our experiments. Firstly, note that we never use the actual MBR judge LLMs to conduct evaluation, and use GPT-4o instead. While GPT-4o could possess biases itself, the benchmarks we employed as well the LLM-as-a-Judge community in general use proprietary LLMs such as GPT-4o as a judge in practice based on the observation that it holds high correlation with human judgments [1][2]. Hence, we emphasize that this is a broader problem for the evaluation community in general, not a specific problem for MBR. Note that we nonetheless take steps to ensure that our gains are real and not the result of overfitting to GPT-4o by conducting a human study on our MBR and BoN decoded outputs (Appendix H). We find that human judges, like GPT-4o, also rate Prometheus MBR outputs more highly than Prometheus BoN or greedy outputs.
>
> Finally, we note that overfitting to the utility metric is a problem known in the MBR literature, and is not specific to LLM judges [3][4][5]. It is also known from prior work in translation that this overfitting problem is worse for BoN decoding (known as Quality Estimation in the translation literature) than for MBR decoding. Assuming that findings from machine translation translate well to instruction-following, one advantage of MBR decoding with LLM judges could be that it in fact overfits comparatively less to the judges' biases than BoN decoding!
>
>
>
> [1] Dubois et al. 2024. Length-Controlled AlpacaEval: A Simple Way to Debias Automatic Evaluators.
>
> [2] Zheng et al. 2024. Judging LLM-as-a-Judge with MT-Bench and Chatbot Arena.
>
> [3] Müller et al. 2021. Understanding the Properties of Minimum Bayes Risk Decoding in Neural Machine Translation.
>
> [4] Freitag et al. 2022. High Quality Rather than High Model Probability: Minimum Bayes Risk Decoding with Neural Metrics.
>
> [5] Fernandes et al. 2022. Quality-Aware Decoding for Neural Machine Translation
>
> ---
>
> *Question 3: The hyperparameter Ncond is set to 12 for generating candidates for DPO but increased to 30 during decoding. How did the authors determine these values, and how do they align with the experiment illustrated in Figure 2?*
>
> Thank you for the question!
>
> From the N_cand curves (right side of Fig. 2), we see that setting N_cand above 10 already recovers most of the gains associated with MBR and BoN inference. In order to balance performance and compute cost, we therefore chose N_cand = 12 for our self-training experiments. As the objective of this section is to demonstrate that MBR distillation is a promising method for self-training, we do not feel that expending considerably more train-time compute to achieve the best possible results is necessary when using a lower N_cand already yields significant gains.
>
> As for our MBR and BoN inference experiments - we could certainly lower N_cand from 30 and still achieve strong performance gains. However, the objective of this section is to understand the full potential of MBR inference performance, so we do feel that using a larger N_cand is necessary. We have modified Section 4.1.2 to explain our selection of N_cand =12 for distillation more clearly. Please see our updated manuscript.

---

> ### Author Response · Authors · 2024-11-29
> **We eagerly await your response**
>
> Dear Reviewer hFLa,
>
> We have responded to your review of our work and have updated our manuscript. We greatly appreciate the time you have taken to help us improve our paper!
>
> With the author-discussion period drawing to a close, we would be grateful if you could respond to our rebuttal - your feedback is crucial to the progress of our work and we would like our paper to be in the best possible shape before the discussion period closes.

---

### Official Review · Reviewer_Nmeb · 2024-11-04

**Soundness:** 3
**Presentation:** 4
**Contribution:** 3
**Rating:** 8
**Confidence:** 3

**Summary:**

This paper proposes an approach for selecting one of N generation hypotheses using a Minimum Bayes Risk (MBR) method. MBR decoding alone results in improved performance. However, as MBR decoding is resource-intensive and may not be practical for real-life applications, an alternative use case—self-training with DPO on preference pairs selected via MBR—also demonstrates performance gains.

**Strengths:**

* The paper explores the use of previously developed approaches—specifically, Minimum Bayes Risk (MBR)—for selecting a generation hypothesis at the decoding stage in LLMs. The MBR-selected hypothesis is treated as the one with the highest average utility according to a specified utility metric. The choice of this approach is well-reasoned, with a clear motivation behind it.

* The experimental setup is solid. The paper first demonstrates improvements using MBR decoding. Given its high computational cost, the paper then explores self-training with MBR-decoded outputs, which also leads to improvements. The evaluation is conducted on two standard benchmarks (Alpaca-Eval 2.0 and MT Bench), with comparisons across different decoding approaches and a variety of utility metrics used in MBR. The discussion of results is well-organized, providing clear and comprehensive takeaways.

**Weaknesses:**

The paper is well-written and easy to follow. The reviewer does not identify any major weaknesses.

**Questions:**

Are there any distinctive linguistic features in MBR-decoded outputs? How does it affect diversity, style, or tone?

---

> ### Author Response · Authors · 2024-11-24
> **Response to Reviewer Nmeb**
>
> Thank you for your review! We’re glad to hear you found the approach well-reasoned, the discussion well-organized and comprehensive, and the writing easy to follow.
>
> ---
> *Question: Are there any distinctive linguistic features in MBR-decoded outputs? How does it affect diversity, style, or tone?*
>
> This is a great question!
>
> One linguistic feature that we analyzed in our original submission is generation length (Appendix A.4 and G.6). We find that MBR-decoded outputs tend to be slightly longer than their greedy and BoN counterparts, although they are still far shorter than **Longest** or **Embedder** decoding.
>
> In response to your question, we conducted some further analysis in search of other distinctive linguistic features. We will include these results in the Appendix of our camera-ready manuscript:
>
> **Formatting**
>
> We used GPT-4o to classify the outputs of Llama-3-8b-Instruct and Llama-3-70b-Instruct on AlpacaEval with greedy, Prometheus BoN, and Prometheus MBR decoding into “Bullet List”, “Numbered List”, “Both” or “Neither” categories, depending on whether a specific kind of list formatting is present in the output. We then computed the percentage of outputs that fall into each category.
>
>
> | Method           | Bullet List | Numbered List | Both  | Neither |
> |-------------------|-------------|---------------|-------|---------|
> | Greedy           | 14.2        | 29.1          | 26.4  | 30.2    |
> | Prometheus BoN   | 13.6        | 29.8          | 26.0  | 30.8    |
> | Prometheus MBR   | 13.9        | 30.1          | 28.1  | 27.8    |
>
> We notice that Prometheus MBR uses list formatting more often than Prometheus BoN and greedy decoding, although the difference is quite small.
>
> **Lexical Diversity**
>
> We calculate the type token ratio (TTR) of the outputs of Llama-3-8b-Instruct and Llama-3-70b-Instruct on AlpacaEval with greedy, Prometheus BoN and Prometheus MBR decoding. This measures the lexical diversity of the resulting outputs (higher => more diverse).
>
> | Method           | TTR   |
> |------------------|-------|
> | Greedy           | 0.514 |
> | Prometheus BoN   | 0.520 |
> | Prometheus MBR   | 0.521 |
>
> Again we notice only very small differences between the outputs. We note that longer generation lengths may be associated with higher lexical diversity, which may act as a confounder.
>
> **Readability**
>
> We compute the Flesch Kincaid readability scores (lower => more readable) of the outputs of Llama-3-8b-Instruct and Llama-3-70b-Instruct on AlpacaEval with greedy, Prometheus BoN and Prometheus MBR decoding.
>
> | Method           | FK Score |
> |------------------|----------|
> | Greedy           | 12.13    |
> | Prometheus BoN   | 12.40    |
> | Prometheus MBR   | 12.24    |
>
> We find there to be little difference in readability scores between the various decoding strategies.
>
> We plan to conduct further follow-up work in the future to better understand the linguistic characteristics of MBR decoding. We would love to hear if you have any ideas for experiments we could try.

---

> > ### Comment · Reviewer_Nmeb · 2024-11-25
> > **Keeping the score**
> >
> > Thank you for your response and for conducting the experiments. I appreciate your efforts. I will maintain my current score, as I believe it is already sufficiently high.

---

> > > ### Author Response · Authors · 2024-11-27
> > > **Response to Reviewer Nmeb**
> > >
> > > Understood. Thank you for your timely response and for your review!

---

### Meta-Review · Area_Chair_C3DL · 2025-01-03

**Metareview:**

This paper introduces a novel application of Minimum Bayes Risk (MBR) decoding to enhance the test-time performance of instruction-following LLMs. LLM judges (including Prometheus2, Llama3, and JudgeLM) are used as reference-based evaluators and select high-quality outputs from a set of candidates generated by Llama 2 & 3 models.  Evaluations on AlpacaEval and MT-Bench demonstrate that MBR decoding with LLM judges significantly outperforms greedy decoding and other reference-free decoding strategies, even with smaller LLMs acting as judges. The paper explores iterative self-training on MBR-decoded outputs. MBR decoding is also combined with self-training (DPO) to yield further gains and overcome the added decoding costs.

There is consensus among the reviewers that the experimental setup and experiments are thorough. The majority of the reviewers agree that the paper is well written, clear to follow, and motivation and results are presented well. One of the concerns raised was the need for comparisons with other decoding methods which is addressed by the authors with speculative decoding results in the rebuttal. Another suggestion was to include a discussion on the computational cost which the authors have included in section 4.2 of the paper and few more details in the rebuttal. Reviewer Z8nY points out that the paper lacks analysis of the dependence of the LLM judge’s quality. The authors rebut with a detailed analysis of this by answering the questions asked by the reviewer. With its solid experimental setup, clear writing, and a rebuttal addressing all key concerns, I am recommending this paper for acceptance.

**Additional Comments On Reviewer Discussion:**

See above.

---

### Decision · Program_Chairs · 2025-01-22

Accept (Spotlight)